# Speculative Sampling For Faster Molecular Dynamics

**Arthur Kosmala** [1 2 3 *] **Stephan Günnemann** [2 3 4] **Meng Gao** [1 †] **Brandon M. Wood** [1 †]

## Abstract

Molecular dynamics (MD) is a key tool for simulating the dynamical behavior of atomic systems. However, MD is inherently serial, which makes it difficult to increase single-system throughput with concurrent compute. To address this, we introduce **L**angevin **S**peculative **D**ynamics (**LSD**), a distributed and model-agnostic speculative sampler for accelerating MD *without adding relative error*. Inspired by speculative methods in language and diffusion modeling, LSD uses a draft model to propose fast simulation steps and verifies them in parallel with a slower target model, applying a transport map from the draft to the target distribution. We extend speculative sampling to second-order Langevin dynamics, derive the achievable speedup as a function of physical parameters, show that LSD generalizes across different systems and draft-target combinations with a 3-9x speedup, and confirm theoretically and empirically that LSD samples trajectories from the same distribution as its target model.

## 1. Introduction

Molecular dynamics (MD) is the standard approach for researchers to study the time evolution of atomic systems across chemistry, biology, and materials science (Frenkel & Smit, 2023). MD simulations use a potential to compute per-atom forces that are then used to numerically integrate Newton's equations of motion. One of the major limitations of MD is that numerical stability requires small individual time steps (Allen & Tildesley, 2017), typically $\sim$ 0.5-1 fs, while many target processes only happen at much longer time scales of 100+ ns, thus putting some out of practical

* Work done at Meta. † Shared corresponding author. [1]FAIR at Meta [2]School of Computation, Information & Technology, Technical University of Munich [3]Munich Data Science Institute [4]Munich Center For Machine Learning. Correspondence to: Brandon M. Wood <bmwood@meta.com>, Meng Gao <rgao@meta.com>, Arthur Kosmala <a.kosmala@tum.de>.

*Proceedings of the $43^{rd}$ International Conference on Machine Learning*, Seoul, South Korea. PMLR 306, 2026. Copyright 2026 by the author(s).

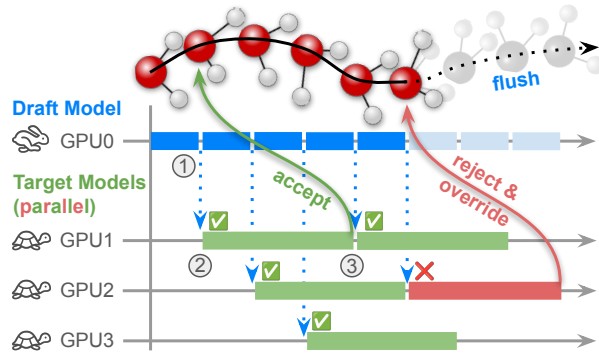

*Figure 1.* LSD algorithm overview: (1) a draft model proposes a stream of fast simulation steps; (2) in parallel, target model instances asynchronously recompute forces on draft steps; (3) as a target model call returns, a stochastic transport map from the draft to the target step distribution either accepts the step (as frequently as possible), or else rejects and overrides it, in which case the draft is rolled back. The result is a parallelism-driven speedup with a draft-independent guarantee of target-distributed trajectories.

reach due to a prohibitive number of serial steps.

One recent development relevant to MD is the introduction of machine learned interatomic potentials (MLIP) (Batzner et al., 2022; Batatia et al., 2023; Wood et al., 2025; Rhodes et al., 2025). MLIPs present an avenue to provide quantum-chemistry level accuracy with linear scaling. However, MLIPs are much slower than classical force fields per force call and thus running a large number of serial steps becomes computationally infeasible.

Large language models with long context windows face an analogous problem at inference time. Specifically, each generated token requires an expensive forward pass conditioned on all prior tokens, so autoregressive decoding is inherently serial. One approach designed to alleviate this problem without a drop in performance is called speculative sampling (Leviathan et al., 2023; Chen et al., 2023; De Bortoli et al., 2025), where a fast draft model is used to make approximate proposals that are validated in parallel by a slower target model. We adapt these ideas to MD and investigate if we can accelerate an inherently serial process while provably preserving a target distribution of physical trajectories.

Similar to other machine learning domains, MLIPs exhibit an accuracy–speed trade-off where more accurate models

tend to be slower at inference time while less accurate models tend to be faster (Wood et al., 2025). This Pareto front naturally provides a pool of candidate draft and target models for speculative sampling.

In this work we introduce Langevin Speculative Dynamics (LSD), a speculative sampler for accelerating MD. LSD uses a fast draft model to propose steps, which are validated in parallel by a target model. Formally, LSD generalizes speculative sampling to the second-order Langevin equation, thus making it applicable to MD. We establish theoretical guarantees that the distribution of LSD-sampled trajectories matches that of the target model for arbitrary draft models. We provide further theoretical analysis of LSD, relating the achievable speedup to a number of factors (draft–target pairing, number of atoms, temperature, etc) and develop speculative error correction which informs current draft predictions by past errors to reduce rejection rates by up to 75%. Finally, we provide empirical results across representative systems that show good agreement between theory and experiment and error-free speedups of up to 9×.

## 2. Foundations

### 2.1. Molecular Dynamics

**MD principles and basic notation.** Molecular dynamics solves the classical laws of motion numerically for a system of interacting atoms. The system's state at any time $t$ is described by the positions $\mathbf{q}_i(t) \in \mathbb{R}^3$, momenta $\mathbf{p}_i(t) \in \mathbb{R}^3$ and masses $m_i \in \mathbb{R}$ of the particles $i \in \{1, \ldots, N\}$. We will often compactly denote the combined vectors of all positions and momenta by $\mathbf{q} \in \mathbb{R}^{3N}, \mathbf{p} \in \mathbb{R}^{3N}$ and use the mass matrix $\mathbf{M} \in \mathbb{R}^{3N \times 3N}$ which is diagonal with entries $m_1, m_1, m_1, \ldots, m_N, m_N, m_N$. The time evolution is prescribed by the total energy (Hamiltonian) function, $H(\mathbf{q}, \mathbf{p}) = \frac{1}{2}\mathbf{p}^{\mathsf{T}}\mathbf{M}^{-1}\mathbf{p} + U(\mathbf{q})$, where $U(\mathbf{q})$ is the *interatomic potential* (modeled, e.g. as an MLIP) and its negative gradient $\mathbf{F}(\mathbf{q}) = -\frac{\partial}{\partial \mathbf{q}}U(\mathbf{q})$ are the forces on the atoms. The system evolves according to Hamilton's equations $\frac{d\mathbf{q}}{dt} = \frac{\partial H}{\partial \mathbf{p}} = \mathbf{M}^{-1}\mathbf{p}$, $\frac{d\mathbf{p}}{dt} = -\frac{\partial H}{\partial \mathbf{q}} = \mathbf{F}(\mathbf{q})$.

**The Langevin thermostat.** Hamilton's equations describe an isolated system that exchanges no energy with its environment. To model more realistic experimental conditions, we frequently want to couple the system to a heat bath (thermostat) that exchanges energy to equilibrate the system at a controlled temperature $T$. A popular choice for modeling the interaction with a heat bath is the *Langevin thermostat*. It couples Hamilton's equations with a friction and a diffusive term, resulting in the following system of (Itô) stochastic differential equations:

$$d\mathbf{q}(t) = \mathbf{M}^{-1}\mathbf{p}(t)\,dt, \tag{1}$$
$$d\mathbf{p}(t) = [\mathbf{F}(\mathbf{q}(t)) - \gamma\mathbf{p}(t)]\,dt + \sqrt{2\gamma k_B T \mathbf{M}}d\mathbf{W}(t),$$

where $\gamma > 0$ is the friction coefficient, $k_B$ is Boltzmann's constant, and $\mathbf{W}(t)$ is a standard Wiener process (Brownian motion). The friction term $-\gamma\mathbf{p}(t)$ dissipates energy while the noise term $\sqrt{2\gamma k_B T \mathbf{M}}d\mathbf{W}(t)$ injects thermal fluctuations. An advantage of the Langevin thermostat is that, under ergodicity and weak regularity assumptions on $U(\mathbf{q})$, the dynamics converge to the correct stationary (Boltzmann) distribution $\pi(\mathbf{q}, \mathbf{p}) \propto \exp(-H(\mathbf{q}, \mathbf{p})/k_B T)$.

**First- vs. second-order Langevin dynamics.** Unlike the Langevin dynamics common to the denoising diffusion model literature, the MD-relevant Eq. (1) is *second-order*. The first-order equation is appropriate for generative modeling as it also converges to a Boltzmann distribution in its position space (Särkkä & Solin, 2019), but Eq. (1) is correct for modeling inertia as it includes positions as well as momenta. Relatedly, while the simple first-order Euler-Maruyama scheme is a popular choice for numerically integrating first-order SDEs in generative modeling (Song et al., 2021), it is uncommon in MD due to its poor energy conservation and limited stability (Leimkuhler & Matthews, 2013). One of our theoretical contributions compared to prior work such as De Bortoli et al. (2025) is therefore to extend the method of speculative sampling to second-order integrators.

**Second-order Langevin numerical integrators.** We can construct second-order integrators by splitting the combined right-hand sides of Eq. (1) into three individually solvable pieces: an (A) term driving the positions by the momenta, a (B) term for deterministic momentum updates from the interatomic forces, and an (O) term for the Ornstein–Uhlenbeck (friction + noise) dynamics on the momenta. Each subproblem (A), (B), (O) can be integrated analytically for a timestep $\Delta t$ or half-step $\frac{\Delta t}{2}$ (cf. Eq. (2)) and composing a full step in different orders (e.g., (ABOBA), (BAOAB), (OB-ABO)) yields splitting schemes with distinct accuracy and stability properties. In the deterministic limit (no (O) term), suitable symmetric orderings reduce to standard symplectic integrators (Verlet, 1967), preserving phase-space structure and energy over long times. Adding the stochastic (O) term to the natural Hamiltonian decomposition (A) + (B) is a natural extension to Langevin dynamics. Additional steps may include center-of-mass fixation, constraint projections or Monte Carlo moves. For a simple exposition in Sec. 3, we focus on the (ABOBA) split as our main example, which performs well for configurational averages (Leimkuhler & Matthews, 2013) and yields the following update rules:

$$
\begin{aligned}
\text{(A)} \quad & \mathbf{q} \leftarrow \mathbf{q} + \Delta t/2\,\mathbf{M}^{-1}\mathbf{p} \\
\text{(B)} \quad & \mathbf{p} \leftarrow \mathbf{p} + \Delta t/2\,\mathbf{F}(\mathbf{q}) \\
\text{(O)} \quad & \mathbf{p} \leftarrow e^{-\gamma\Delta t}\,\mathbf{p} + \boldsymbol{\Sigma}^{\frac{1}{2}}\,\boldsymbol{\xi} \\
\text{(B)} \quad & \mathbf{p} \leftarrow \mathbf{p} + \Delta t/2\,\mathbf{F}(\mathbf{q}) \\
\text{(A)} \quad & \mathbf{q} \leftarrow \mathbf{q} + \Delta t/2\,\mathbf{M}^{-1}\mathbf{p},
\end{aligned} \tag{2}
$$

where $\boldsymbol{\xi} \leftarrow \mathcal{N}(\mathbf{0}, \mathbb{I})$ and $\boldsymbol{\Sigma} = \mathbf{M}k_B T\left(1 - e^{-2\gamma\Delta t}\right)$ (3)

## 2.2. Speculative Sampling

**Molecular dynamics meets speculative sampling.** Let us now introduce a notation that abstracts away MD specifics and will instead let us focus on the connection to speculative sampling. As $\mathbf{q}$ and $\mathbf{p}$ together make up the state of a second-order Langevin time step, we group them into a vector

$$x_n := (\mathbf{q}_n, \mathbf{p}_n) \in \mathbb{R}^{6N} =: \mathcal{X}. \tag{4}$$

Integrator schemes like Eq. (2) effectively define a randomized update $x_{n-1} \mapsto x_n$ that samples from a conditional density, $x_n \leftarrow P(\cdot|x_{n-1})$. The Markovianity is specific to our MD setting, but the content of this section applies to general autoregression if we simply swap out $P(\cdot|x_{n-1})$ for $P(\cdot|x_0, \ldots, x_{n-1})$ and let $\mathcal{X}$ be any discrete or continuous state space instead of $\mathbb{R}^{6N}$. We call $P(\cdot|x_{n-1})$ the *target model*, to be distinguished from the *target force field* (e.g., an MLIP) that computes $\mathbf{F}$ in Eq. (2). While getting $\mathbf{F}$ (done once per integrator step) dominates the cost of sampling $x_n \leftarrow P(\cdot|x_{n-1})$, the target model really is a combination of both a target force field and an integrator scheme.

**Fast drafting, parallel verification.** Running Langevin MD directly with the target model ties our throughput to the rate at which we can sample from $P(\cdot|\cdot)$ step after step. To resolve this bottleneck, speculative sampling shifts the serial workload onto a faster *draft* model $Q(\cdot|\cdot)$ that proposes draft steps $y_n := (\tilde{\mathbf{q}}_n, \tilde{\mathbf{p}}_n) \in \mathbb{R}^{6N} =: \mathcal{Y}$. In the case of MD, we realize $Q(\cdot|\cdot)$ by swapping out the target forces $\mathbf{F}$ for draft forces $\tilde{\mathbf{F}}$ (obtained from a faster force field) in an integrator scheme such as Eq. (2). Crucially, a subsequent verification of draft steps by the target model $P(\cdot|\cdot)$ can now be done on multiple finished draft steps *in parallel*. While most prior works such as Leviathan et al. (2023) alternate between rolling out a fixed number of $L$ new draft steps (while all the target model instances sit idle) and synchronously applying $P(\cdot|\cdot)$ afterwards to these $L$ steps, LSD pipelines this into an asynchronous algorithm without a pre-specified lookahead integer $L$, see App. Sec. D for details.

**Probabilistic verification of draft steps.** After we have materialized $Q(\cdot|y_{n-1})$ and $P(\cdot|y_{n-1})$ on an input draft step $y_{n-1}$, the former at *draft time* and the latter at a subsequent walltime that we call *verification time*, we transport the draft samples $y_n$ onto verified samples $x_n$ by some randomized map. This map must be crafted in a way that transforms steps $y_n$ from $Q(\cdot|y_{n-1})$ onto new steps $x_n$ from $P(\cdot|y_{n-1})$. Many such maps exist, but ours should *also* maximize the probability that $x_n = y_n$. The unfortunate event $x_n \neq y_n$ (a *rejection*) stalls our drafting progress at $x_n$, as all later steps already drafted or verified by the current walltime assume stale inputs. We have no choice but to flush them and restart drafting from the just overridden state $x_n$. On the other hand, doing so guarantees recursively that all kept verified outputs $x_n$ come from a $P(\cdot|x_{n-1})$ where the input $x_{n-1}$ is *also*

verified (not flushed). By induction in $k$, the joint distribution $P(x_1, \ldots, x_n|x_0) = \prod_{k=1}^{n} P(x_k|x_{k-1})$ of simulation trajectories starting at a fixed initial condition at $x_0$ until the current step $x_n$ is thus identical to what sequential sampling with the target model would have produced.

**Coupling and maximality criteria.** Formally, we define the verification of draft samples as a randomized function

$$\text{VERIFY}(\cdot\,;\cdot \mid P, Q, n)\colon \mathcal{Y} \times \mathcal{U} \to \mathcal{X} \tag{5}$$

that outputs a simulation state $x_n \in \mathcal{X}$ based on the drafted input state $y_n \in \mathcal{Y}$ and internal randomness $u \in \mathcal{U}$ drawn from a random variable $U$, such as $U \sim \text{Uniform}([0,1])$. If we denote $X_n, Y_n$ for the random variables of which $x_n, y_n$ are drawn, our requirements of the previous paragraph read

$$Y_n \sim Q(\cdot|y_{n-1}) \implies \tag{6}$$
$$X_n = \text{VERIFY}(Y_n; U \mid P, Q, n) \sim P(\cdot|y_{n-1}),$$
$$\mathbb{P}(X_n = Y_n) \geq \mathbb{P}(\tilde{X}_n = \tilde{Y}_n) \tag{7}$$
$$\forall (\tilde{X}_n, \tilde{Y}_n)\colon \tilde{X}_n \sim P(\cdot|y_{n-1}) \wedge \tilde{Y}_n \sim Q(\cdot|y_{n-1}).$$

We call Eq. (6) the *coupling* and Eq. (7) the *maximality* property. The objects characterized by Eqs. (6) and (7) are known as *maximal couplings* (Lindvall, 2002) and have previously been linked to speculative sampling by De Bortoli et al. (2025). We reframe parts of this section in the equivalent language of maximal couplings in App. Sec. A.

## 3. Langevin Speculative Dynamics

### 3.1. Verification of MD Steps

At its very core, developing a speculative sampling method for MD comes down to finding a randomized transport map that fulfills the coupling and maximality criteria Eqs. (6) and (7) between MD-specific target and draft distributions $P(\cdot|y_{n-1})$, $Q(\cdot|y_{n-1})$. As laid out in Sec. 2.2, our draft steps are $y_{n-1} = (\tilde{\mathbf{q}}_{n-1}, \tilde{\mathbf{p}}_{n-1})$ and sampling from $P(\cdot|y_{n-1})$ / $Q(\cdot|y_{n-1})$ means stepping through an integrator scheme like (ABOBA) (Eq. (2)) with forces $\mathbf{F}$ / $\tilde{\mathbf{F}}$ obtained from a target / draft force field at input $y_{n-1}$.

**Verification of momentum updates.** Together, the three middle (BOB) steps of Eq. (2) with a draft force field $\tilde{\mathbf{F}}$ constitute a Gaussian draw for the new draft momentum $\tilde{\mathbf{p}}_n$,

$$\tilde{\mathbf{p}}_n \leftarrow \mathcal{N}(\cdot\,; \langle \tilde{\mathbf{p}}_n \rangle, \mathbf{\Sigma}) \tag{8}$$
$$\langle \tilde{\mathbf{p}}_n \rangle = e^{-\gamma\Delta t}\tilde{\mathbf{p}}_{n-1} + \left(1 + e^{-\gamma\Delta t}\right)(\Delta t/2)\,\tilde{\mathbf{F}}_{\mathbf{n}},$$
$$\mathbf{\Sigma} = \mathbf{M}k_B T\left(1 - e^{-2\gamma\Delta t}\right), \tag{BOB}$$

Crucially, when we repeat these steps with the target forces $\mathbf{F}$ at verification time, the *only* difference that arises is in the mean updated momentum $\langle \mathbf{p_n} \rangle \neq \langle \tilde{\mathbf{p}}_n \rangle$, while the covariance $\mathbf{\Sigma}$ does not depend on the force field. Before we

turn towards coupling two full integrator steps in $\mathbb{R}^{6N}$, it therefore is helpful to understand how it is done for the middle two $3N$-dimensional (BOB) momentum Gaussians with different means but identical and diagonal covariance matrix $\boldsymbol{\Sigma}$. We solve this through *reflection-maximal couplings* (Bou-Rabee et al., 2020; De Bortoli et al., 2025), visualized for a single dimension in Figs. 2a to 2c. Given the previous, updated and mean updated draft momenta $\tilde{\mathbf{p}}_{n-1}, \tilde{\mathbf{p}}_n, \langle\tilde{\mathbf{p}}_n\rangle$ (obtained from $\tilde{\mathbf{F}}$ at drafting time) and the mean updated target momentum $\langle\mathbf{p}_n\rangle$ (obtained later from $\mathbf{F}$ at verification time), the idea is to accept draft momenta based on the target/draft likelihood ratio and reflect rejected momenta (Mahalanobis-)normally to the equal-likelihood hyperplane. With $\mathbf{z} := \boldsymbol{\Sigma}^{-\frac{1}{2}}(\tilde{\mathbf{p}}_n - \langle\tilde{\mathbf{p}}_n\rangle)$ and the normal $\boldsymbol{\delta} := \boldsymbol{\Sigma}^{-\frac{1}{2}}(\langle\tilde{\mathbf{p}}_n\rangle - \langle\mathbf{p}_n\rangle)$, REFLECTIONVERIFY reads

$$\mathbf{p}_n = \text{REFLECTIONVERIFY}(\tilde{\mathbf{p}}_n; u \mid \langle\mathbf{p}_n\rangle, \langle\tilde{\mathbf{p}}_n\rangle)$$

$$= \begin{cases} \tilde{\mathbf{p}}_n, & \text{if } u \leq \min\left\{1, \frac{\mathcal{N}(\tilde{\mathbf{p}}_n; \langle\mathbf{p}_n\rangle, \boldsymbol{\Sigma})}{\mathcal{N}(\tilde{\mathbf{p}}_n; \langle\tilde{\mathbf{p}}_n\rangle, \boldsymbol{\Sigma})}\right\}, \\ \langle\mathbf{p}_n\rangle + \boldsymbol{\Sigma}^{\frac{1}{2}}\left(\text{Id} - 2(\boldsymbol{\delta}^\top\boldsymbol{\delta})^{-1}\boldsymbol{\delta}\boldsymbol{\delta}^\top\right)\mathbf{z}, & \text{else,} \end{cases}$$

where $u \leftarrow \text{Uniform}([0,1])$. (9)

As proven by Bou-Rabee et al. (2020) in a general setting, the coupling and maximality requirements Eqs. (6) and (7) with the two Gaussians $\mathcal{N}(\tilde{\mathbf{p}}_n; \langle\mathbf{p}_n\rangle, \boldsymbol{\Sigma}), \mathcal{N}(\tilde{\mathbf{p}}_n; \langle\tilde{\mathbf{p}}_n\rangle, \boldsymbol{\Sigma})$ both hold and the rejection rate $\beta_n = \mathbb{P}(\mathbf{p}_n \neq \tilde{\mathbf{p}}_n)$ is

$$\beta_n = \int \max\{0, \mathcal{N}(\tilde{\mathbf{p}}; \langle\mathbf{p}_n\rangle, \boldsymbol{\Sigma}) - \mathcal{N}(\tilde{\mathbf{p}}; \langle\tilde{\mathbf{p}}_n\rangle, \boldsymbol{\Sigma})\}d\tilde{\mathbf{p}}$$

$$= \text{erf}\left(\|\boldsymbol{\delta}\|/\sqrt{8}\right), \quad (10)$$

where $\|\boldsymbol{\delta}\| = \|\boldsymbol{\Sigma}^{-\frac{1}{2}}(\langle\tilde{\mathbf{p}}_n\rangle - \langle\mathbf{p}_n\rangle)\|$ is the Mahalanobis distance of the updated draft/target momentum means and erf the error function. In Fig. 2c, $\beta_n$ amounts to the red area.

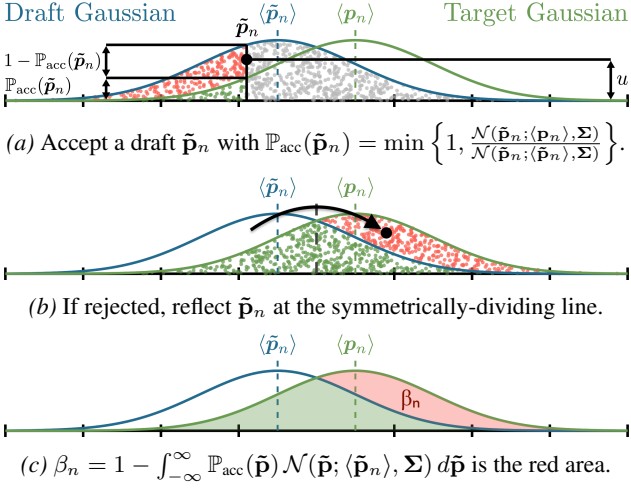

Draft Gaussian $\quad \tilde{\mathbf{p}}_n \quad \langle\tilde{\mathbf{p}}_n\rangle \quad \langle\mathbf{p}_n\rangle \quad$ Target Gaussian

$1 - \mathbb{P}_{\text{acc}}(\tilde{\mathbf{p}}_n)$
$\mathbb{P}_{\text{acc}}(\tilde{\mathbf{p}}_n)$
$u$

*(a)* Accept a draft $\tilde{\mathbf{p}}_n$ with $\mathbb{P}_{\text{acc}}(\tilde{\mathbf{p}}_n) = \min\left\{1, \frac{\mathcal{N}(\tilde{\mathbf{p}}_n; \langle\mathbf{p}_n\rangle, \boldsymbol{\Sigma})}{\mathcal{N}(\tilde{\mathbf{p}}_n; \langle\tilde{\mathbf{p}}_n\rangle, \boldsymbol{\Sigma})}\right\}$.

$\langle\tilde{\mathbf{p}}_n\rangle \quad \langle\mathbf{p}_n\rangle$

*(b)* If rejected, reflect $\tilde{\mathbf{p}}_n$ at the symmetrically-dividing line.

$\langle\tilde{\mathbf{p}}_n\rangle \quad \langle\mathbf{p}_n\rangle$

$\beta_n$

*(c)* $\beta_n = 1 - \int_{-\infty}^{\infty} \mathbb{P}_{\text{acc}}(\tilde{\mathbf{p}}) \mathcal{N}(\tilde{\mathbf{p}}; \langle\tilde{\mathbf{p}}_n\rangle, \boldsymbol{\Sigma}) d\tilde{\mathbf{p}}$ is the red area.

*Figure 2.* REFLECTIONVERIFY for two 1D Gaussians.

**Verification of a full integrator step.** The (BOB) part of Eq. (2) does not alter positions, so the map acting as RE-

FLECTIONVERIFY on the momenta and as the identity on the positions trivially embeds our (BOB) momentum coupling into the full $\mathbb{R}^{6N}$ state space. The question is how we can derive a full (ABOBA) coupling from our known coupling between (BOB) parts. Intuitively, if we have aligned distributions after (BOB), they should remain aligned when we apply (A) to the verified outcomes as (A) does not differ from draft to target. Moreover, even if we knew a coupling that aligns the distributions after (A), no such map can have better acceptance rates, as the invertible post-processing through (A) does not affect overlap between distributions. We formalize this intuition in Thm. 3.1:

**Theorem 3.1** (Pre-/post-processing). *Assume that the target and draft distributions $P, Q$ are decomposable into the form*

$$P(\cdot|y_{n-1}) = g_* P'(\cdot|f(y_{n-1}))$$
$$Q(\cdot|y_{n-1}) = g_* Q'(\cdot|f(y_{n-1}))$$

*with functions $f, g$, conditional distributions $P', Q'$, and the pushforward operator $*$. Let $Y_n' \sim Q'(\cdot|f(y_{n-1}))$ record the draft state before the pushforward under $g$ was applied.*

*If a randomized map $\text{VERIFY}(\cdot; \cdot \mid P', Q', n)$, defined via Eq. (5), obeys the coupling property Eq. (6) for $P', Q'$, then*

$$X_n := g \circ \text{VERIFY}(Y_n'; U \mid P', Q', n) \sim P(\cdot|y_{n-1}).$$

*If $\text{VERIFY}(\cdot; \cdot \mid P', Q', n)$ moreover satisfies maximality (Eq. (7)) for $P', Q'$ and if $g$ is invertible, then no coupling of $P, Q$ has an acceptance rate higher than $\mathbb{P}(X_n = Y_n)$.*

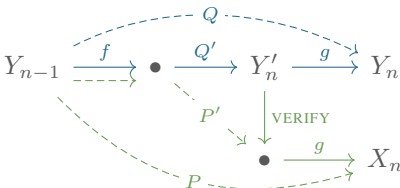

*Figure 3.* Transition kernel diagram illustrating Thm. 3.1. Fully tracing the blue vs. green paths samples a step from the draft vs. target distribution. The paths taken by LSD are drawn solid.

As a direct consequence of Thm. 3.1, applying REFLECTION-VERIFY to the draft momenta after the (BOB) step and reapplying (A) to the verified outputs couples the draft and target (ABOBA) steps and has the optimal acceptance rate, as RE-FLECTIONVERIFY satisfies maximality and (A) is invertible. We prove Thm. 3.1 in App. Sec. B. Unlike (ABOBA), some integrator schemes may include non-invertible post-processing steps such as center-of-mass fixation or constraint projections. Through Thm. 3.1, the coupling property still applies to such schemes as long as all force-dependent steps can be grouped into a contiguous sampling block for which a coupling map is known. We generalize our findings to a different integrator scheme, (OBABO), in App. Sec. F. Moreover, we provide the full pipelined LSD algorithm and an end-to-end correctness proof in App. Sec. D.

### 3.2. Speedup and Optimal Resource Bounds

Let us now study the algorithm-level characteristics of LSD.

**Optimization through pipelining.** To achieve optimal performance in practice, we pursue a pipelined multi-device solution instead of the synchronous Alg. 1 used in seminal language works like Leviathan et al. (2023). Our target models run on their own devices and act as an asynchronous pool that verifies draft steps by the next-idle model. A verifier protocol aggregates target model outputs and immediately rolls back the draft after a rejection to prevent stale draft inputs (cf. Fig. 1). The pipelined variant is also simpler to analyze as it no longer depends on a lookup integer $L$ like Alg. 1. We discuss our detailed reasons for choosing a pipelined approach and prove its correctness in App. Sec. D.

**Walltime speedup analysis.** We derive an implementation-agnostic upper bound on the expected speedup from our method. It depends on only two inputs: an upper rejection bound on all steps, and the draft/target cost fraction. We discuss in App. Sec. D how this result simplifies and improves the non-pipelined bound of (De Bortoli et al., 2025).

**Proposition 3.1** (Upper speedup bound). *Let $r > 0$ lower-bound the rejection rates for all draft steps and let $c > 0$ be the draft-to-target cost ratio (assumed constant). For i.i.d. rejections, the expected speedup is upper-bounded by $\frac{1}{c+r}$.*

*Proof.* The LSD walltime cost of a single length-$L$ accept streak including the rejection (Fig. 1) is $L\Delta t_{\text{fast}} + \Delta t_{\text{slow}}$ (due to the overhead of verifying the rejected step), whereas the serial target model cost is $L\Delta t_{\text{slow}}$. In expectation,

$$\mathbb{E}\left(\frac{L\Delta t_{\text{slow}}}{L\Delta t_{\text{fast}} + \Delta t_{\text{slow}}}\right) = \mathbb{E}\left(\frac{1}{c + L^{-1}}\right) \leq \frac{1}{c + \mathbb{E}(L)^{-1}},$$

where we used Jensen's inequality with a concave $f(x) = 1/(c+x^{-1})$ given $c > 0$ in the last step. By assumption, we can lower-bound $\mathbb{E}(L)$ by a geometric expectation, $\mathbb{E}(L) \leq \sum_{l=1}^{\infty}(1-r)^{l-1}rl = \frac{1}{r}$, which implies the statement. $\square$

**Optimal resource allocation.** Prop. 3.1 assumes enough target model devices exist to always serve new draft steps immediately. Assuming that a single target model offers a stable throughput of $\mu$ steps / second, a pool of $N_T$ such models can serve draft steps at a combined rate $N_T\mu$. By conservation of flow, a draft throughput of $\lambda$ requires $N_T \geq \lceil\frac{\lambda}{\mu}\rceil = \lceil\frac{1}{c}\rceil$ target models to avoid waiting times. As steps cannot be served faster than their rate of arrival, allocating even more target models just adds an average of $N_T - \frac{1}{c}$ idle target models. Mild overallocation can still be a helpful buffer against fluctuations in true arrival and service rates.

### 3.3. Prediction of the Speedup and Rejection Rate

Prop. 3.1 in itself makes no direct speedup predictions yet as its upper rejection bound $r$ is undetermined. We also still

need to understand how rejection rates depend on different simulation sizes, temperatures, frictions, and MD timesteps. We find an insightful analysis under two simplifications:

**Assumption 1.** Rejections are i.i.d. events happening at a constant, "time-averaged rate" $0 \leq \langle\beta\rangle \leq 1$ for all steps.

**Assumption 2.** In a simulation box repeating a group of equal atoms $G \propto N$ times, every group adds an equal and $G$-independent amount to the total draft-target error.

Assump. 1 imposes homogeneity in time and helps us derive an empirical speedup formula, while Assump. 2, amounting to homogeneity in space, lets us model the dependence of force errors (and therefore rejection rates) on system size.

**Empirical speedup formula.** Under Assump. 1, already central to the seminal analysis of Leviathan et al. (2023), the unknown bound $r$ in Prop. 3.1 instead becomes the constant rate $\langle\beta\rangle$ that can be consistently estimated from data as the ratio $\frac{R(K)}{K} \to \langle\beta\rangle$ of rejected vs. total steps for $K \to \infty$. The result is an empirically testable formula,

$$\text{speedup} \lesssim \frac{1}{c + \langle\beta\rangle} \tag{11}$$

where the inequality is due to added overheads (e.g., communication) not modeled in the bound of Prop. 3.1.

**Speedup formula interpretation.** Eq. (11) has a compellingly simple structure: the cost fraction and mean rejection rate, which represent the Pareto tradeoff between draft throughput and accuracy, have symmetric effects. Improving one characteristic far beyond the other shows diminishing returns as speedups remain dominated by the other, larger parameter. Therefore, optimizing draft-target combinations should strike a balance between both factors. We embed our speedup measurements in Sec. 5, Fig. 5 in a contour plot of this expected theoretical landscape, confirming the tightness of Eq. (11) for a majority of cases.

**Modeling of the rejection rate.** In Sec. C, we use Assumps. 1 and 2 to derive a semi-empirical relationship for the mean rejection rate $\langle\beta\rangle$ depending on the atom count $N$, temperature $T$, timestep $\Delta t$ and friction timescale $\tau = \frac{1}{\gamma}$:

$$\exists \varepsilon > 0\colon \langle\beta\rangle(N, \tau, \Delta t, T) \approx \text{erf}\left((N\tau\Delta t)^{\frac{1}{2}}T^{-\frac{1}{2}}\varepsilon\right) \tag{12}$$

where the empirical constant $\varepsilon$ captures the draft-target error per atom and is *independent* of $N, \tau, \Delta t$ and $T$. In practice, if one fixes the draft and target model (and therefore $\varepsilon$), measuring a rate $\langle\beta\rangle_*$ once for an input choice $(N, \tau, \Delta t, T)_*$ allows inverting Eq. (12) to estimate $\varepsilon$ and extrapolate $\langle\beta\rangle$ to all other choices of $(N, \tau, \Delta t, T)$. Intuitively,

- higher temperatures $T$ / larger frictions $\gamma$ (or smaller $\tau = \gamma^{-1}$) increase thermal noise, making the Gaussians in Fig. 2c less distinguishable due to higher $\Sigma$,

- longer timesteps $\Delta t$ increase the Gaussian distance $\|\delta\|$ at fixed $\Sigma$, making the Gaussians more distinguishable,

- higher atom count $N$ increases Gaussian dimension, increasing $\|\delta\|$ if per-atom errors $\|\delta\|/\sqrt{N}$ are fixed.

We emphasize that Assumps. 1 and 2 reflect strong approximations about the system and therefore think of Eq. (12) primarily as an informative model. While it is in fact highly accurate for the copper system of Sec. 5, Fig. 4, complicating factors like spatial / temporal inhomogeneity or long-range interactions may impact its quantitative accuracy.

**Speculative error correction (EC).** In regimes where Eq. (12) dictates high rejection rates, the obvious measure is to align the draft and target, i.e., lower the constant $\varepsilon$. One way to do this is by learning from past errors. Let $\tilde{\mathbf{F}}_n$ be the bare force output of the draft model at the current step $n$, and $\mathbf{F}_{n-k}$, $\tilde{\mathbf{F}}_{n-k}$ be the bare draft and target outputs at the most recent verified step $n - k$. If the error $\Delta\mathbf{F}_{n-k} = \mathbf{F}_{n-k} - \tilde{\mathbf{F}}_{n-k}$ evolves more slowly than the two forces, we can approximate $\mathbf{F}_n \approx \tilde{\mathbf{F}}_n + \Delta\mathbf{F}_{n-k}$, essentially speculating that the known past error approximates the current error. EC substitutes bare draft outputs $\tilde{\mathbf{F}}_n$ by this expression and also caches the raw outputs $\tilde{\mathbf{F}}_n$ for future comparison to keep track of the most recent historical error. We provide a more detailed, algorithm-level discussion of EC in App. Sec. E and explore another related strategy to improve rejection rates in App. Sec. H.

## 4. Related Work

**Speculative sampling and related methods.** Speculative sampling in its probabilistic form was concurrently introduced for language model decoding by Leviathan et al. (2023); Chen et al. (2023), and later generalized to image denoising diffusion models by De Bortoli et al. (2025). However, it is predated by conceptually similar approaches in the Hamiltonian Monte Carlo (HMC) community, including the introduction of reflection-maximal couplings by Bou-Rabee et al. (2020) that crucially underlies this work as well as De Bortoli et al. (2025). For molecular simulations, a close relative of LSD is hybrid Monte Carlo (Duane et al., 1987), which accepts or rejects steps via an energy-based Metropolis-Hastings criterion and has been used to correct MLIP proposals with concurrently running DFT evaluations by Nagai et al. (2020). Unlike LSD, the acceptance criterion of HMC-based methods requires an explicit energy function (Hamiltonian), whereas LSD operates purely on forces and therefore readily admits a non-conservative draft model. The guarantees also differ: LSD recovers a target product distribution of integrator steps, while HMC ensures asymptotically exact Boltzmann sampling.

**Accelerating MLIP driven MD.** Many recent works have proposed methods to accelerate MLIP driven MD with sim-

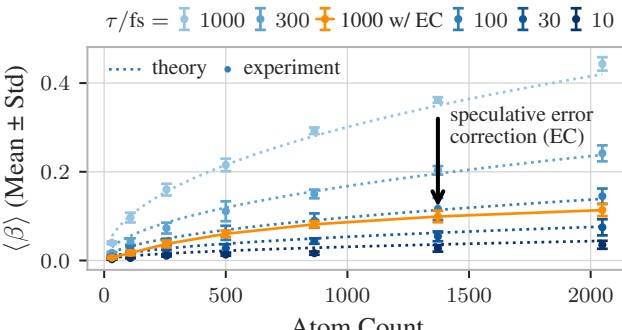

*Figure 4.* The mean Orb-v3-direct + UMA-S rejection rates for FCC copper at varying atom count $N$ and friction timescale $\tau$ agree strongly with Eq. (12) fitted to $(N = 500, \tau = 1\text{ps})$. EC makes LSD viable at $\tau = 1\text{ps}$ by strongly reducing rejections.

ilar motivations as ours. Example strategies include predicting large segments of a trajectory at once through large timesteps (Bigi et al., 2025a; Klein et al., 2023; Viguera Diez et al., 2025), reusing embeddings from previous timesteps (Schaaf et al., 2024), distillation of faster models (Amin et al., 2025; Cattin et al., 2026), or using a hierarchy of interaction cutoffs (Fu et al., 2023), which is closely related multiple time-step integration (Tuckerman et al., 1992). A common issue with these methods is that they are typically lossy and may rely on additional hyperparameters, whereas LSD provides guarantees to match the target model trajectory distribution. Additionally, some of these techniques could be used as a fast draft model in an LSD sampler that is then verified by the target model.

## 5. Experiments

### 5.1. Predicted and Real Rejection Rates

The achievable speedup in Eq. (11) crucially depends on rejection rates, so we validate our model of their dependence on system and simulation parameters. In (Fig. 4), our benchmark system is FCC copper with sizes of $32 \leq N \leq 2048$ atoms at temperature $T = 1500K$ and timestep $\Delta t = 1\text{fs}$. The draft is `Orb-v3-direct-20-omat` (Rhodes et al., 2025), one of the most lightweight pretrained MLIPs at the time of writing. Our target is `UMA-S` (Wood et al., 2025), a universal MLIP with excellent accuracy and generalization.

**Agreement with theory.** In Fig. 4, we plot the mean rejection rates $\langle\beta\rangle$, estimated from the ratios $\frac{R(K)}{K}$ of rejected vs. total time steps for trajectory segments of length $K = 1000$, at varying atom counts $N$ and friction timescales $\tau$. To assess the match with theory, we can invert Eq. (12) on the single pair $(N = 500, \tau = 1\text{ps})$ to get an estimate for the draft-target-associated constant $\varepsilon$. Using Eq. (12) with this $\varepsilon$ accurately models $\langle\beta\rangle$ for other values of $(N, \tau)$.

**Improvement from speculative error correction.** We now consider the highest friction timescale of $\tau = 1\text{ps}$. Running LSD naively at this setting quickly degrades $\langle\beta\rangle$

with growing atom counts above practical values. However, EC combats this by lowering $\langle\beta\rangle$ far enough to make $\tau = 1$ps LSD simulations practically viable. We further explore the impact of EC on rejection rates in App. Sec. K.

### 5.2. Predicted and Real Speedups

We now study predicted vs. real speedups of different draft-target combinations on FCC copper ($T = 1500K$, $\tau = 1$ps) and how Eq. (11) informs the tradeoffs in picking a draft.

**Draft and target models.** As target models, we use two MLIPs: `UMA-S` from Sec. 5.1 as well as the costlier `UMA-M`. As LSD works with any draft (not just MLIPs), we test effective medium theory (`EMT`), a fast classical force field for metals, as well as two draft MLIPs: `Orb-v3-direct-20-omat` and the slightly costlier, but more accurate `UMA-tiny-direct` adapted from Fu et al. (2025) (see Sec. N).

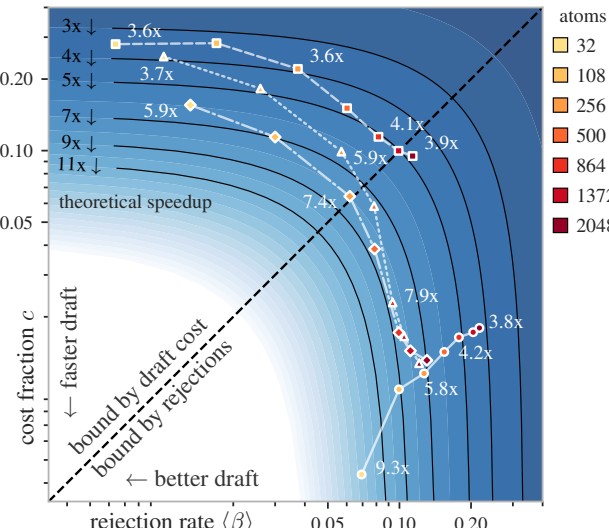

*Figure 5.* Each marker represents cost fractions $c$ and mean rejection rates $\langle\beta\rangle$ for a draft/target combination on FCC copper at varying atom count $N$. True speedups are upper-bounded by Eq. (11) which is *only* a function of $(c, \langle\beta\rangle)$ and plotted as the contour landscape, as opposed to the true speedups (white labels) which also depend on other inputs, e.g. communication overhead. We orient GPU counts at the throughput optimum from Sec. 3.2.

**Pareto analysis.** In Fig. 5, we compare the theoretical speedup predicted by Eq. (11) (black contours) as a function of draft / target cost fractions $c$ and mean rejection rates $\langle\beta\rangle$ to the measured true speedup (white text labels). As expected, Eq. (11) is an upper bound. It tends to be tighter with the slow `UMA-M` target, relative to which the LSD overhead is small, while the gap to theory is largest for the `UMA-S` / `EMT` combination, which is fastest in absolute throughput. Two antagonistic trends are clear from Fig. 5:

- **Rejections increase with atom count.** This matches the theory from Sec. 3 along with the prior subsection.

- **Draft MLIP cost fractions decrease with atom count.** Perhaps less obviously, this is because smaller draft MLIPs benefit longer from near-flat initial scaling (ideal parallelism) than large target MLIPs on the same GPU at growing atom counts. While both models naturally slow down, the target slows earlier and faster.

The winning trend determines if speedups tend to decrease or, perhaps counterintuitively, increase with system size as observed with MLIP drafts in Fig. 5. Notably, the second effect is associated with draft models leveraging GPU parallelism and does not apply to the CPU-based `EMT` model.

**Regime crossover.** We find it interesting to note that the two MLIP combinations involving the slow `UMA-M` target model cross from a firmly cost- into a rejection-bound regime, making the decision whether to focus on a fast or accurate draft model entirely dependent on the simulated atom count.

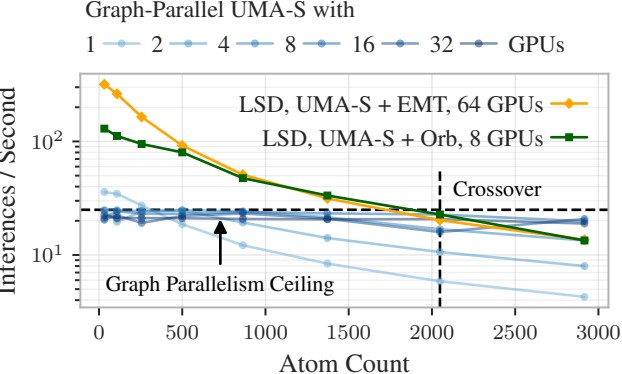

*Figure 6.* LSD and graph parallelism throughput vs. atom count. The Graph Parallelism (GP) ceiling is caused by a fixed compute overhead ( 20ms) specific to `UMA-S`. For small atom count, the GP speedup is compromised by the ceiling and LSD shows much better performance. GP outperforms LSD at larger atom counts.

**Comparison to graph parallelism.** An obvious question is how much we can gain from LSD compared to spatial domain decomposition approaches like graph parallelism (GP) (Sriram et al., 2022). In order for GP to be effective, one needs to overcome distributed communication bottlenecks (especially for message-passing networks) and fixed costs (see App. Sec. L) to achieve strong scaling characteristics. This can be very challenging and require model-specific optimizations for many MLIP architectures (Park et al., 2024). LSD is insensitive to such target model internals as it parallelizes target model calls *end-to-end, in time*. However, the maximum LSD speedup is fundamentally capped (Eq. (11)) by the rejection rate that increases with growing system size. This leads to a crossover, illustrated for `UMA-S` in Fig. 6: LSD performs an order of magnitude better than graph parallelism for small $N$ and trails GP for large $N$. Both methods

can be combined to add concurrency over time and space.

### 5.3. Preservation of Target Distribution

A core advantage of using LSD as a speedup technique is the preservation of the target model distribution of trajectories. In principle (and if no force slack is employed), this is mathematically guaranteed by Thm. 3.1, but we test it in multiple experiments to assert the correctness of LSD, including further parity tests in App. Sec. J and M.1.

**Non-conservative heating for bulk water.** Due to its guarantees, we can use LSD to correct a non-conservative (direct-force) draft model by an energy-conserving target model. Like Bigi et al. (2025b), we find that non-conservative MLIPs cause excess heating and test if LSD can correct the problem. To this aim, we simulate 250ps trajectories of bulk water ($\tau = 1$ps and $N = 300$) and record the kinetic temperature $T_{\mathrm{kin}} = \frac{2K}{Nk_B}$, a quantity proportional to the kinetic energy $K$. While it fluctuates due to Langevin noise, its average over a long trajectory should match our thermostat setting of $T = 300K$ *if the model conserves energy*. In Tab. 1 we list the mean excess kinetic temperatures $\langle T_{\mathrm{kin}} \rangle - 300K$ obtained for the conservative `UMA-S`, non-conservative `UMA-tiny-direct-omol` as well as the LSD combination of both models. As expected, `UMA-S` shows almost no

*Table 1.* Mean excess heating for a conservative target model, a non-conservative draft model and their LSD combination.

| Force Field | UMA-S | LSD | UMA-tiny-direct |
|---|---|---|---|
| $\langle T_{\mathrm{kin}} \rangle - 300K$ | $1.0_{\pm 0.9}$ | $1.1_{\pm 0.8}$ | $42.8_{\pm 0.7}$ |

mean excess heating (except for a residue that we attribute to the numerical integrator), while `UMA-tiny-direct` misses the target temperature by more than 40K. Consistently with its guarantees, LSD corrects this effect entirely.

**Lithium diffusion in LGPS.** To assert target model parity for a kinetic observable, we check if LSD reproduces target statistics for Lithium diffusion in the $Li_{10}GeP_2S_{12}$ (LGPS)

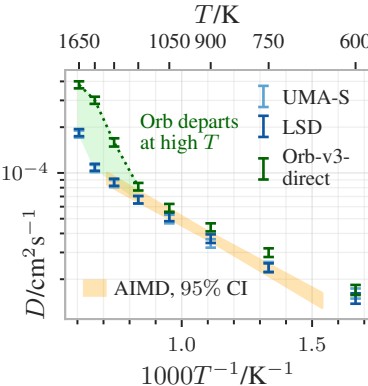

superionic conductor (Kamaya et al., 2011). We compute Li self-diffusivities, $\lim_{t\to\infty} \frac{\mathrm{MSD}(t)}{6t}$, by Bayesian linear regression of Lithium mean squared displacements of a trajectory against time (McCluskey et al., 2024) for `UMA-S`, `Orb-v3-direct-20-omat` as well as their LSD combination (cf. Fig. 7).

*Figure 7.* Li diffusivities w/ 95% CI vs. (inverse) temperatures for an `UMA` target, `Orb` draft and their LSD combination.

As an ab-initio reference, we also plot the Arrhenius relationship (95% confidence interval) fitted to the AIMD trajectories obtained between 650K and 1400K by López et al. (2024). While the target model accurately reproduces the ab-initio data in its covered regime, the draft tends to overestimate diffusivity. Both MLIPs deviate from the expected Arrhenius relationship for high $T$, but the draft departs much more significantly. Consistently with its guarantees, LSD matches its target model diffusivities within sampling error.

**High-dimensional parity test**. So far, we have tested for parity of several important physical observables, all of which have corresponded to long-time averages taken over low-dimensional marginals of the full trajectory distribution.

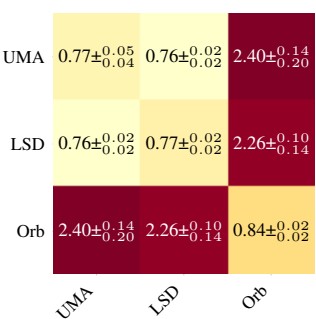

We now test the other extreme: parity of the high-dimensional joint distributions over short trajectories. By sampling 1000 of such 10-step trajectories for 108-atom copper using UMA-S, ORB-v3-DIRECT-20-OMAT and their LSD combination and treating each set of trajectories as a high-dimensional point cloud, we estimate the distance between their underlying distributions in a reproducing kernel Hilbert space by the maximum

*Figure 8.* MMD-estimated Gaussian kernel distances (median $\sigma$, amplified by factor $10^3$) between distributions over copper trajectories, treating each trajectory as a high-dimensional point, sampled with UMA-S, ORB and LSD.

mean discrepancy (MMD) statistic (Gretton et al., 2012) applied to pairs of such point clouds. For background on MMD and details of our setup, we refer to App. Sec. G. Fig. 8 shows that the LSD vs. UMA MMD is not higher than the MMD between UMA and itself at different random seeds, while ORB clearly samples from a different distribution of trajectories.

## 6. Conclusion and Limitations

We developed LSD, a novel speculative sampler that addresses the serial bottleneck of molecular dynamics through a combination of fast drafting and concurrent verification. We proved parity with the target distribution, analyzed rejection rates and speedup, and empirically validated our predictions. Our results demonstrate meaningful, error-free speedups on realistic systems; however, performance is system-size-dependent. For systems with $> \mathcal{O}(10^3)$ atoms, rejection rates become limiting and other forms of parallelism may be preferred, while for smaller systems speedups are often bound by the cost of the draft model. Furthermore, LSD focuses on throughput under the assumption of suffi-

cient compute, which may not always be practical. In future work, distillation or online inference-time learning could improve draft accuracy and extend the applicability of LSD to larger atom counts, and specialized draft models could improve cost fractions.

## Code Availability

We plan to release a reference implementation of LSD on `https://github.com/facebookresearch/LSD` soon after publication.

## Impact Statement

Our work aims to accelerate scientific discovery in chemistry and materials science using MLIP-based simulations; however, potential misuses exist. The method presented in this work improves the simulation lengths obtainable with an existing target model, and we made sure that all models used for this work were trained on datasets designed for applications beneficial to society to mitigate these risks.

## Acknowledgements

The authors thank Juno Nam for the several useful discussions about MD settings and his help in the preparation of starting structures, as well as Filippo Bigi and Michele Ceriotti for sharing their insights about splitting schemes and bulk water MD. A.K. also thanks Vitalia Khamenya for her warm encouragement throughout all stages of this work.

A.K. acknowledges support by the Munich Data Science Institute (MDSI) at Technical University of Munich (TUM) via the Linde/MDSI Doctoral Fellowship.

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

## A. Maximal Couplings

**From random maps to conditional distributions.** As an alternative and equivalent perspective to the random map formulation used throughout the main body of this work, we can note down the map VERIFY as the operation sampling a value $x_n$ of the verified random variable $X_n$ from a conditional distribution $\Pi_{X_n|Y_n}(\cdot|y_n)$,

$$\text{VERIFY}(y_n \mid P, Q) =: x_n \leftarrow X_n \mid Y_n \sim \Pi_{X_n|Y_n}(\cdot|y_n), \tag{13}$$

To recover the coupling and maximality properties Eqs. (6) and (7) in this new language, we first remark that $\Pi_{X_n|Y_n}(\cdot|y_n)$ needs to be chosen such that it marginalizes to

$$X_n \sim \int \Pi_{X_n|Y_n}(\cdot|y_n)Q(y_n|y_{n-1})dy_n \stackrel{!}{=} P(\cdot|y_{n-1}). \tag{14}$$

Instead of taking the conditional map $\Pi_{X_n|Y_n}(\cdot|y_n)$ into the centerpoint, the route taken by this work due to its closeness to implementation, others such as De Bortoli et al. (2025) have instead treated the product density $\Pi_{X_n,Y_n}(x_n, y_n) := \Pi_{X_n|Y_n}(x_n|y_n)Q(y_n|y_{n-1})$ as their central object. Note that, when we use the term 'density' and integral notation, we implicitly refer to a suitable integration measure for $X_n$. This unified treatment also includes all discrete probability mass functions (under the counting measure) as well as real generalized "densities" involving $\delta$ distributions (under linear combinations of the Lebesgue and Dirac measures).

**Maximal couplings.** The product density is a coupling of $X_n, Y_n$ due to the way it marginalizes: $\int \Pi_{X_n,Y_n}(x_n, y_n)dy_n = P(\cdot|y_{n-1})$ by Eq. (14), and $\int \Pi_{X_n,Y_n}(x_n, y_n)dx_n = Q(\cdot|y_{n-1})$ simply due to normalization. In this formulation, the acceptance rate $\alpha := P(X_n = Y_n) = 1 - \beta$ becomes

$$\alpha = \int \mathbb{1}_{\{x \in \mathcal{X}, y \in \mathcal{Y}|x=y\}} \Pi_{X_n,Y_n}(X_n, Y_n) d^2 x_n y_n, \tag{15}$$

with $\mathbb{1}$ denoting the indicator function, and essentially corresponds to the total probability mass concentrated on the diagonal of the product density $\Pi_{X_n,Y_n}$. Our maximality property Eq. (7) then means that out of all product densities that marginalize to be a coupling of $X_n, Y_n$, a *maximal* coupling should concentrate as much probability as possible on its diagonal. For further general treatment of maximal couplings we refer to Lindvall (2002) and for an alternative perspective on the map REFLECTIONVERIFY as a reflection-maximal coupling, we refer to Bou-Rabee et al. (2020); De Bortoli et al. (2025).

## B. Proof of Thm. 3.1

**Theorem 3.1** (Pre-/post-processing)**.** *Assume that the target and draft distributions $P, Q$ are decomposable into the form*

$$P(\cdot|y_{n-1}) = g_* P'(\cdot|f(y_{n-1}))$$
$$Q(\cdot|y_{n-1}) = g_* Q'(\cdot|f(y_{n-1}))$$

*with functions $f, g$, conditional distributions $P', Q'$, and the pushforward operator $*$. Let $Y'_n \sim Q'(\cdot|f(y_{n-1}))$ record the draft state before the pushforward under $g$ was applied.*

*If a randomized map* VERIFY$(\cdot; \cdot \mid P', Q', n)$*, defined via Eq. (5), obeys the coupling property Eq. (6) for $P', Q'$, then*

$$X_n := g \circ \text{VERIFY}(Y'_n; U \mid P', Q', n) \sim P(\cdot|y_{n-1}).$$

*If* VERIFY$(\cdot; \cdot \mid P', Q', n)$ *moreover satisfies maximality (Eq. (7)) for $P', Q'$ and if $g$ is invertible, then no coupling of $P, Q$ has an acceptance rate higher than $\mathbb{P}(X_n = Y_n)$.*

*Proof.* We first prove the first part of the statement, conservation of the coupling property. By assumption, VERIFY fulfills the coupling property Eq. (6) with respect to $Q', P'$, which directly implies

$$Y'_n \sim Q'(\cdot|f(y_{n-1})) \implies \text{VERIFY}(Y'_n; U \mid P', Q', n) \sim P'(\cdot|f(y_{n-1})).$$

By the definition of the pushforward operation $*$ and the decomposition property of $P$ assumed in the theorem,

$$X_n = [g \circ \text{VERIFY}](Y'_n; U \mid P', Q', n) \sim g_* P'(\cdot|f(y_{n-1})) = P(\cdot|y_{n-1}), \tag{16}$$

which proves the first part of the theorem statement.

For the second part, we also assume that VERIFY fulfills the maximality property Eq. (7) and that $g$ is invertible. The proof proceeds by contradiction. Assume that a map VERIFY$'$ exists that fulfills the coupling property Eq. (6) for $P, Q$ and has a higher acceptance rate than $\mathbb{P}(X_n = Y_n)$, i.e.

$$Y_n \sim Q(\cdot|y_{n-1}) \implies \tilde{X}_n := \text{VERIFY}'(Y_n; U' \mid P, Q, n) \sim P(\cdot|y_{n-1}), \qquad \text{(contradiction assumption 1)}$$

$$\mathbb{P}(\tilde{X}_n = Y_n) > \mathbb{P}(X_n = Y_n). \qquad \text{(contradiction assumption 2)}$$

The following construction would be possible:

$g$ by assumption has an inverse $g^{-1}$, so $\tilde{X}'_n = [g^{-1} \circ \text{VERIFY}' \circ g](Y'_n; U' \mid P, Q, n)$ is well-defined. The event $\tilde{X}_n = Y_n$ (i.e., VERIFY' acting as the identity) clearly implies the event $\tilde{X}'_n = Y'_n$. Thus,

$$\mathbb{P}(\tilde{X}'_n = Y'_n) \geq \mathbb{P}(\tilde{X}_n = Y_n) > \mathbb{P}(X_n = Y_n), \qquad (17)$$

where the second inequality is contradiction assumption 2.

Let us finally define $X'_n := \text{VERIFY}(Y'_n; U \mid P', Q', n)$. Since, by construction, $X_n = g(X'_n)$ and $Y_n = g(Y'_n)$, $(X'_n = Y'_n) \implies (X_n = Y_n)$ and therefore $\mathbb{P}(X_n = Y_n) \geq \mathbb{P}(X'_n = Y'_n)$. Together with Eq. (17), this implies

$$\mathbb{P}(\tilde{X}'_n = Y'_n) > \mathbb{P}(X'_n = Y'_n) \qquad (18)$$

On the other hand, $[g^{-1} \circ \text{VERIFY}' \circ g]$ fulfills the coupling property for $P', Q'$ as, by contradiction assumption 1, $\tilde{X}_n = [\text{VERIFY}' \circ g](Y'_n; U' \mid P, Q, n) = \text{VERIFY}'(Y_n; U' \mid P, Q, n) \sim P(\cdot|y_{n-1})$ and, therefore, by the assumed decomposition of $P$ and the definition of the pushforward operation $*$,

$$Y'_n \sim Q'(\cdot|f(y_{n-1})) \implies \tilde{X}'_n = g^{-1}(\tilde{X}_n) = [g^{-1} \circ \text{VERIFY}' \circ g](Y'_n; U' \mid P, Q, n) \sim P'(\cdot|f(y_{n-1})). \qquad (19)$$

Eqs. (18) and (19) would imply the existence of a coupling of $P', Q'$ with higher acceptance rate than VERIFY, contradicting the assumed maximality of VERIFY and thus proving the statement. $\qquad \square$

## C. Derivation of the rejection rate scaling formula

Let us derive Eq. (12) from the following two simplifying assumptions:

**Assumption 1.** Rejections are i.i.d. events happening at a constant, "time-averaged rate" $0 \leq \langle \beta \rangle \leq 1$ for all steps.

**Assumption 2.** In a simulation box repeating a group of equal atoms $G \propto N$ times, every group adds an equal and $G$-independent amount to the total draft-target error.

By Eq. (10), the Mahalanobis distance $\|\boldsymbol{\delta}\|$ between the draft/target Gaussians of the momentum update determines the rejection rate of a verification step. The definitions of the updated momentum mean $\langle \tilde{\mathbf{p}}_n \rangle$, covariance $\boldsymbol{\Sigma}$ (Eq. (3)) and friction timescale $\tau = \frac{1}{\gamma} \gg \Delta t$ lead to the expression

$$\|\boldsymbol{\delta}\| = \left( \frac{\tau \Delta t}{2k_B T} \right)^{\frac{1}{2}} \|\boldsymbol{M}^{-\frac{1}{2}} \Delta \mathbf{F}\| + \mathcal{O}\left( (\Delta t/\tau)^2 \right) \qquad (20)$$

where $\Delta \mathbf{F} \in \mathbb{R}^{3N} = \tilde{\mathbf{F}} - \mathbf{F}$ is the draft-target force error. Following Assump. 1, we neglect the variance of rejections in time and instead simply assume a constant, time-independent error. Assump. 2 then implies that each of the $G$ groups adds an equal and $G$-independent amount to the mass-scaled error term $\|\boldsymbol{M}^{-\frac{1}{2}} \Delta \mathbf{F}\|$. $G$-independence means the system can be scaled up or down without significant change to the force errors per atom group, which holds in particular if the draft and target model are local.

Assump. 2 justifies the scaling ansatz $\|\boldsymbol{M}^{-\frac{1}{2}} \Delta \mathbf{F}\| = \sqrt{G} \tilde{\varepsilon}$ where $\tilde{\varepsilon}$ is the draft-target error per unit cell and the $\sqrt{G}$ scaling is due to orthogonality of per-atom forces in the all-atom space $\mathbb{R}^{3N}$. Therefore, we get the formula

$$\exists \varepsilon > 0 : \langle \beta \rangle (N, \tau, \Delta t, T) \approx \text{erf}\left( (N\tau\Delta t)^{\frac{1}{2}} T^{-\frac{1}{2}} \varepsilon \right) \qquad (21)$$

where we absorbed all terms that do not depend on $N, \tau, \Delta t$ and $T$ into the empirical constant $\varepsilon$. As $\varepsilon$ is proportional to the draft-target error per unit cell $\tilde{\varepsilon}$, it is fixed for a given draft-target combination and needs to be determined empirically.

---

**Algorithm 1** Speculative Sampling Step (Synchronous)

---

**input** Lookahead integer $L$, initial sequence $x_{0:n}$, draft model $Q(\cdot|\cdot)$, target model $P(\cdot|\cdot)$.

    **for** $n \leq j < n + L$ **do**
        Get $Q(\cdot|y_{0:j})$
        Sample $y_j \leftarrow Q(\cdot|y_{0:j})$
    **end for**
    **in parallel for** $n \leq j < n + L$ **do**
        Get $P(\cdot|y_{0:j})$
        Sample $x_j \leftarrow \text{VERIFY}(y_j; u|P, Q)$
    **end for**
    $l \leftarrow \min\{j \mid n \leq j < n + L, \, x_j \neq y_j\}$
    $x_{0:l+1} \leftarrow x_{0:n} + [x_n, \ldots, x_l]$
**output** $x_{0:l+1}$

---

# D. Discussion of the pipelined verification approach

## D.1. Motivation for pipelining

**Synchronous vs. pipelined verification.** Seminal works from language modeling like Leviathan et al. (2023); Chen et al. (2023) and most follow-up works essentially use variants of the basic synchronous Alg. 1 above (in contrast to MD, LLM autoregression is non-Markovian and hence we adapted the notation with $y_{0:n} := (y_0, \ldots y_{n-1})$). It alternates between the drafting of $L$ new steps while all target model instances are running idle, and synchronous parallel verification while the draft model is idle. Our method, in contrast, is a producer-consumer-type approach where draft steps get served as soon as a target model instance is idle. A verifier protocol handles the asynchronous target model outputs and rolls back the draft model in case of a rejection to ensure that it operates on valid inputs. In our context, this choice adds efficiency through pipeline parallelism and also admits a qualitatively simpler analysis of the achievable speedup, which no longer depends on a lookup integer $L$.

**Domain-specific problem constraints.** Many academic setups in the LLM domain place the draft and target model on the same GPU, which takes full pipeline parallelism out of consideration due to the resulting contention. In contrast, pipelining is a natural optimization strategy for multi-device verification and has indeed been explored in this specific context by prior LLM work (McDanel, 2025). For molecular simulations, multi-device verification (and along with it, optimization through pipelining) becomes the sensible default: evaluating state-of-the-art MLIPs on a single input system often consumes the memory of a single modern GPU already at modest sizes of $\mathcal{O}(10^3)$ to $\mathcal{O}(10^4)$ atoms (Wood et al., 2025). Moreover, inter-GPU communication overheads are often small compared to the typical pretrained MLIP query times ($\mathcal{O}(10-100)$ms), which further offsets the costs of a distributed multi-GPU setup.

## D.2. Ensuring end-to-end correctness of the pipelined verification protocol

Our goal for the rest of Sec. D.2 is to verify formally that the pipelined LSD protocol Alg. 2 strictly samples trajectories from the joint distribution defined by the target model, even if the asynchronous target model calls return their verification results out of submission order. Before we discuss any of the internal bookkeeping in Alg. 2, let us first define a recursive black-box condition on its outputs that is equivalent to correct joint sampling, but more natural to prove for Alg. 2.

**Definition D.1** (Recursive correctness). Let $\mathcal{T} = (x_0, x_1, \ldots, x_K)$ denote a trajectory of $K + 1 \in \mathbb{N}$ MD states starting at initial state $x_0$ and $P(\cdot \mid \cdot)$ denote the single-step conditional distribution of the target model. We call $x_n \in \mathcal{T}$ `verified` if

- $n = 0$, or if
- $n > 0$ and we can certify that $x_n$ was drawn from a random variable $X_n \sim P(\cdot \mid x_{n-1})$, where $x_{n-1}$ is `verified`.

If we can assert that each state $x_n$ in the output of Alg. 2 is always `verified` as in Def. D.1, this implies (by induction in $n$) that its outputs $\mathcal{T}$ come from the $x_0$-conditioned target joint distribution, i.e., $P(x_K, \ldots, x_1|x_0) = \prod_{n=1}^{K} P(x_n|x_{n-1})$.

**LSD algorithm and comments.** For a simplified exposition to Alg. 2, our comments below draw a conceptual outline for how its PROCESS_NEW_RETURNS procedure ensures a consistent verifier state despite out-of-order target model returns.

- Due to draft rollbacks after rejections, the pipeline might (at multiple consecutive wall times) propose several draft state updates $y \mapsto y'$ for the same simulation time. We therefore represent draft steps as tuples $(y', \texttt{id})$ of a proposed new state and a unique, immutable, ascending integer key $\texttt{id}$ that identifies this step throughout its lifetime.

- After submitting a new draft step to a target model for verification, we register it in a buffer of `eligible_pending` steps. This keeps track of all steps that are awaiting a target model result *and* have not been flushed by a rejection.

- Asynchronously during verification, a target model maps the updated states $y'$ to reflection-coupled states $x'$ according to the random map of Thm. 3.1 and returns the result to the main process. As discussed in Sec. 2.2 and proven in App. Sec. B, this mapping guarantees that $x'$ is a valid sample of the *target*-distributed random variable $X' \sim P(\cdot \,|\, y)$, where $y$ is the draft state based on which the original draft update $y'$ was computed.

- Each time after the draft emits a new step and submits it to an idle target model, we call a routine that gathers all newly returned results from the pool of target models without blocking. If no target model returned since our last check, we proceed directly to the next draft call, else we iterate over all returned steps as follows:

    - * If a returned step is still found in `eligible_pending`, we remove it from `eligible_pending` and add it to a buffer of `conditionally_verified` ids. Note that this happens *regardless* of acceptance ($x' = y'$) or rejection ($x' \neq y'$).
      * Else, it means that the step has previously been flushed by an upstream rejection (see below). We abort here, i.e. ignore the target model result on the flushed step, and proceed to handle other newly returned steps.
    - If $x' \neq y'$, we remove all steps with larger $\texttt{id}$ from `eligible_pending` and `conditionally_verified` and roll back the current draft state to $x'$ (i.e., flush the downstream pipeline) whereas $\texttt{id}$ just keeps counting up.
    - We then check if there is at least one draft step in `conditionally_verified` without pending predecessors. If true, we pop the earliest such step and append it to a buffer of `verified` steps. We repeat this inner loop until no more states can be added to `verified`, then proceed to handle other newly returned steps.

The role of `conditionally_verified` is therefore to cache out-of-order returns. Crucially, its downstream content is flushed as soon as we learn of an upstream rejection, and no further stale content can be added as `eligible_pending` is also immediately flushed and the draft is rolled back to the last valid state before it can propose a new step. No step is moved from `conditionally_verified` to `verified` unless we can exclude that any predecessor state has been, or could still be rejected. We formalize these insights while we proceed with the end-to-end correctness proof for Alg. 2 below.

**Formal verification of Alg. 2.** We now assert that all elements in the output $\texttt{V}$ of Alg. 2 satisfy the recursive correctness condition Def. D.1. First, note that (after their initialization) the step buffers EP, CV, V, which together define the *verifier state*, are modified only during the call of PROCESS_NEW_RETURNS and the preceding line, both of which run synchronously in the main event loop. None of the parallel workers running VERIFY_STEP modifies the verifier state, so race conditions between the main and worker processes cannot occur. Instead, all program behaviors that act on the verifier state serialize into a sequence of state transitions in the main event loop that we can partition based on the $\texttt{id}$ value at which they occur.

Since PROCESS_NEW_RETURNS always first removes a step from one buffer before (possibly) adding it to another and no $\texttt{id}$ is assigned more than once, it is clear that the five propositions $E, C, V, F, U$ defined below are exhaustive and mutually exclusive $\forall \texttt{id} \in \mathbb{N}_0 \, \forall \texttt{id}' \in \mathbb{N}_0$ ($\texttt{id}$ is the *current* event loop iteration and $\texttt{id}'$ points to *any completed or future* step):

1. $E \Leftrightarrow \exists (y'', \texttt{id}'') \in \texttt{EP} : \texttt{id}'' = \texttt{id}'$,

2. $C \Leftrightarrow \exists (x'', \texttt{id}'') \in \texttt{CV} : \texttt{id}'' = \texttt{id}'$,

3. $V \Leftrightarrow \exists (x'', \texttt{id}'') \in \texttt{V} : \texttt{id}'' = \texttt{id}'$,

4. $F \Leftrightarrow \texttt{id}' < \texttt{id} \wedge \neg(E \vee C \vee V)$,

5. $U \Leftrightarrow \texttt{id}' \geq \texttt{id} \wedge \neg(E \vee C \vee V)$.

Qualitatively, these propositions enumerate the five respective cases where the $\texttt{id}'$-indexed draft step is eligible-pending, conditionally-verified, verified, flushed, or not (yet) created while the event loop is in iteration $\texttt{id}$. Their exhaustive-and-mutually-exclusive property implies that we can identify the set of all verifier states Alg. 2 with the set $\{E, C, V, F, U\}^{\mathbb{N}}$.

The program starts in the state $(V, U, U, \dots)$: the initial condition $x_0$, formally "generated" by the "step" with $id = 0$, starts as verified, while all next steps are not yet created. Each main event loop iteration executes transitions $\{E, C, V, F, U\}^{\mathbb{N}} \to \{E, C, V, F, U\}^{\mathbb{N}}$. Even though the exact transitions taken in each iteration clearly cannot be predicted from the current verifier state alone (they depend on the results *and* the precise return timing of the concurrent VERIFY_STEP procedures), we show next that Alg. 2 constrains the structure of possible transitions sufficiently well to guarantee two statements:

**Theorem D.2** (End-to-end correctness). *Alg. 2 satisfies two properties:*

1. *After every completed event loop iteration* `id`*, the ordered values of the* `V` *buffer are a sequence of states* $(x_0, \dots, x_n)$ *satisfying the recursive correctness condition of Def. D.1,*

2. *Alg. 2 eventually halts if we can guarantee that every event loop iteration finishes after a finite time.*

**Corollary D.3.** *Thm. D.2 clearly implies that Alg. 2 always returns an output which is moreover (see Def. D.1) distributed according to the target joint distribution of trajectories.*

*Proof.* We will abbreviate "Proposition $X$ is true for step `id`$'$ in event loop iteration `id`" (with $X \in \{E, C, V, F, U\}$) by the shorthand notation $X(\texttt{id}, \texttt{id}')$. Each event loop iteration `id` executes a sequence of zero, one, or multiple `id`$'$ state transitions, which we will abbreviate using the notation $X(\texttt{id}, \texttt{id}') \mapsto Y(\texttt{id}, \texttt{id}')$. Strictly speaking, Alg. 2 implements the removal and addition of an `id`$'$ from one buffer into another in two directly subsequent program instructions, and hence moving `id`$'$ implies a transient transition into the $F$ state for one instruction duration. We may however always consider the two instructions as one atomic moving operation without loss of generality, as we have already established that races between the main and worker processes do not occur on the verifier state. Finally, we can define (due to the indexing of draft steps by a unique `id`$'$):

- $y'\texttt{[id}']:= [(y'', \texttt{id}'') \in \texttt{EP}: \texttt{id}'' = \texttt{id}'][0,0]$ if $E(\texttt{id}, \texttt{id}')$,

- $x'\texttt{[id}']:= [(x'', \texttt{id}'') \in \texttt{CV}: \texttt{id}'' = \texttt{id}'][0,0]$ if $C(\texttt{id}, \texttt{id}')$,

- $x'\texttt{[id}']:= [(x'', \texttt{id}'') \in \texttt{V}: \texttt{id}'' = \texttt{id}'][0,0]$ if $V(\texttt{id}, \texttt{id}')$.

As a first step towards the proof, we note that Alg. 2 admits no transitions of the form $V(\texttt{id}, \texttt{id}') \mapsto X(\texttt{id}, \texttt{id}')$, and admits transitions $X(\texttt{id}, \texttt{id}') \mapsto V(\texttt{id}, \texttt{id}')$ only if $X = C$. We can therefore focus on the conditions under which Alg. 2 executes a transition $C(\texttt{id}, \texttt{id}') \mapsto V(\texttt{id}, \texttt{id}')$, as this is the only way in which the state of the `V` buffer can change during an event loop iteration. The following Lemma D.4 is helpful:

**Lemma D.4.** *After any event loop iteration of Alg. 2 with counter* $\texttt{id} \in \mathbb{N}_0$*, the following is true for any* $\texttt{id}' \in \mathbb{N}$*:*

$$C(\texttt{id}, \texttt{id}') \implies (\exists \texttt{id}'' < \texttt{id}': V(\texttt{id}, \texttt{id}'') \wedge x'\texttt{[id}'] \sim P(\cdot \,|\, x''\texttt{[id}''])) $$
$$\vee (\exists \texttt{id}'' < \texttt{id}': C(\texttt{id}, \texttt{id}'') \wedge x'\texttt{[id}'] \sim P(\cdot \,|\, x''\texttt{[id}''])) $$
$$\vee (\exists \texttt{id}'' < \texttt{id}': E(\texttt{id}, \texttt{id}'') \wedge x'\texttt{[id}'] \sim P(\cdot \,|\, y''\texttt{[id}''])) $$

*where* $P(\cdot \,|\, \cdot)$ *is the single-step conditional distribution of the target model.*

*Proof of Lemma D.4.* Suppose $C(\texttt{id}, \texttt{id}')$ while the event loop counter is `id`. By Thm. 3.1, VERIFY_STEP guarantees $x'\texttt{[id}'] \sim P(\cdot \,|\, y)$, where $y$ is the draft state passed when step `id`$'$ was submitted. It suffices to identify $y$ with $x''\texttt{[id}'']$ for some `id`$''$ satisfying $V(\texttt{id}, \texttt{id}'')$ or $C(\texttt{id}, \texttt{id}'')$, or with $y''\texttt{[id}'']$ for some `id`$''$ satisfying $E(\texttt{id}, \texttt{id}'')$.

**Identifying** $y$**.** In Alg. 2, the draft state $y$ is assigned at three locations: (a) initialization sets $y \leftarrow x_0 = x''\texttt{[0]}$, with step 0 in `V`; (b) the line after submission sets $y \leftarrow y'$, the draft proposal of the step just added to `EP`; and (c) rejection in PROCESS_NEW_RETURNS sets $y \leftarrow x'$, the verified state of the rejected step, which resides in `CV` (added before the rejection check and retained by the flush, which keeps `CV` entries with $\texttt{id} \le \texttt{id}_r$). Writing `id`$''$ for the step whose state $y$ was set to, we have $\texttt{id}'' < \texttt{id}'$ in each case (since $y$ was last assigned during the iteration preceding the one that created step `id`$'$), and step `id`$''$ was in `V`, `EP`, or `CV` respectively at that time.

**Step** `id`$''$ **was not flushed by iteration** `id`**.** Since $\texttt{id}'' < \texttt{id}'$ and $C(\texttt{id}, \texttt{id}')$ implies step `id`$'$ was created, step `id`$''$ was also created, ruling out $U(\texttt{id}, \texttt{id}'')$. Suppose for contradiction that $F(\texttt{id}, \texttt{id}'')$ holds, meaning step `id`$''$ was flushed

by a rejection of some step $\text{id}_r$. The flush boundaries (EP and CV each retain $\text{id} \leq \text{id}_r$) imply $\text{id}_r < \text{id}''$. Since $\text{id}' > \text{id}'' \geq \text{id}_r$, the same flush also removed step $\text{id}'$ from either EP or CV. Once removed from EP, the **continue** guard prevents step $\text{id}'$ from ever re-entering CV. Hence $C(\text{id}, \text{id}')$ cannot hold, leading to a contradiction.

**Matching the disjuncts.** At iteration $\text{id}$, step $\text{id}''$ is in $\{E, C, V\}$. If $y$ was set via (a) or (c), then $y = x''[\text{id}'']$ and step $\text{id}''$ is still in state $C$ or $V$ (if it was not in $V$ from the outset, the only transition possible is $C(\text{id}_v, \text{id}'') \mapsto V(\text{id}_v, \text{id}'')$ in some iteration with $\text{id}'' < \text{id}_v \leq \text{id}'$ since we already excluded a flush), so then the first or second disjunct holds. If $y$ was set via (b), then $y = y''[\text{id}'']$ and step $\text{id}''$ was initially in $E$. If it remains in $E$, the third disjunct holds. Otherwise, step $\text{id}''$ has transitioned from $E$ to $C$ (or further to $V$), meaning its verification returned. Had the step been rejected ($x''[\text{id}''] \neq y''[\text{id}'']$), the resulting flush would remove step $\text{id}'$ as shown above, contradicting $C(\text{id}, \text{id}')$. Therefore the step was accepted: $x''[\text{id}''] = y''[\text{id}''] = y$, and the first or second disjunct applies. $\square$

We now use Lemma D.4 to complete the proof of Thm. D.2.

**Statement 1 (Recursive correctness).** As noted above, the only transition affecting V is $C(\text{id}, \text{id}') \mapsto V(\text{id}, \text{id}')$, executed by the **while** loop in PROCESS_NEW_RETURNS: step $\text{id}'$ is popped from CV and appended to V only when $\text{id}' = \min(\text{CV}[:,1])$ and $\text{id}' < \min(\text{EP}[:,1] + [\infty])$. Consider the moment when this transition occurs. By the reasoning in the proof of Lemma D.4, we have $x'[\text{id}'] \sim P(\cdot \mid y)$ where $y$ is the draft state of some step $\text{id}'' < \text{id}'$ that satisfies one of the three cases in Lemma D.4. Assuming a transition $C(\text{id}, \text{id}') \mapsto V(\text{id}, \text{id}')$ excludes two of them:

- $\text{id}''$ is not in CV, since $\text{id}'' < \text{id}' = \min(\text{CV}[:,1])$,

- $\text{id}''$ is not in EP, since $\text{id}'' < \text{id}' < \min(\text{EP}[:,1] + [\infty])$.

Therefore $V(\text{id}, \text{id}'')$ must hold, so by Lemma D.4 $x'[\text{id}'] \sim P(\cdot \mid x''[\text{id}''])$ with $x''[\text{id}'']$ already in V.

It remains to show that $\text{id}''$ is the immediate predecessor of $\text{id}'$ in the ordered V buffer. First, note that the transition $C(\text{id}, \text{id}') \mapsto V(\text{id}, \text{id}')$ always appends $\text{id}'$ at the end of V in strictly increasing $\text{id}$ order: any step $\text{id}'''$ with $\text{id}''' \leq \max(\text{V}[:,1])$ satisfies $V(\text{id}, \text{id}''')$ or $F(\text{id}, \text{id}''')$ (having previously undergone $C \mapsto V$ or been flushed), hence $\neg C(\text{id}, \text{id}''')$. We show $\text{id}'' = \max(\text{V}[:,1])$, i.e., $\neg V(\text{id}, \text{id}''')$ for all $\text{id}'''$ with $\text{id}'' < \text{id}''' < \text{id}'$. In cases (a) and (b) of the proof of Lemma D.4, $\text{id}'' \in \{0, \text{id}' - 1\}$, so no such $\text{id}'''$ exists. In case (c), $\text{id}''$ is a rejected step: $V(\text{id}, \text{id}''')$ would require a prior transition $C(\text{id}, \text{id}''') \mapsto V(\text{id}, \text{id}''')$, which is blocked by its **while** loop guard as long as $E(\text{id}, \text{id}'')$ holds (since $\text{id}'' < \text{id}'''$). Step $\text{id}''$ satisfies $E(\text{id}, \text{id}'')$ until its verification returns; upon return, the transition $E(\text{id}, \text{id}'') \mapsto C(\text{id}, \text{id}'')$ triggers a rejection flush (since $x''[\text{id}''] \neq y''[\text{id}'']$ in case (c)), which causes $C(\text{id}, \text{id}''') \mapsto F(\text{id}, \text{id}''')$ or $E(\text{id}, \text{id}''') \mapsto F(\text{id}, \text{id}''')$ before step $\text{id}'''$ can undergo $C \mapsto V$.

Therefore, each transition $C(\text{id}, \text{id}') \mapsto V(\text{id}, \text{id}')$ appends an element satisfying $x'[\text{id}'] \sim P(\cdot \mid x''[\text{id}''])$, where $x''[\text{id}'']$ is the last element of V before appending and is `verified` by the inductive hypothesis, which shows that the recursive correctness condition Def. D.1 (and by extension, correct sampling from the target joint distribution) is satisfied.

**Statement 2 (Halting).** We show that $|\text{V}|$ increases by at least one after finitely many event loop iterations, which implies termination since $|\text{V}|$ must grow from 1 to $K + 1$. As long as $|\text{V}| < K + 1$, each iteration emits a new draft step, so after this event loop iteration, V has either grown immediately by the new step or there exists an $\text{id}'$ with $E(\text{id}, \text{id}') \vee C(\text{id}, \text{id}')$. In the second case, let $m = \min\{\text{id}' : E(\text{id}, \text{id}') \vee C(\text{id}, \text{id}')\}$. For all $\text{id}''' < m$, only $V(\text{id}, \text{id}''')$ or $F(\text{id}, \text{id}''')$ can hold (by minimality of $m$, and since all such steps have been created, ruling out $U$). Steps satisfying $V$ or $F$ cannot trigger new rejections ($V$ steps were already processed; returns from $F$ steps are discarded by the **continue** guard). Therefore the transitions $E(\text{id}, m) \mapsto F(\text{id}, m)$ and $C(\text{id}, m) \mapsto F(\text{id}, m)$ cannot occur as any flush requires a prior rejection.

If $C(\text{id}, m)$: all steps with $\text{id}' \leq m$ satisfy $\neg E(\text{id}, \text{id}')$ (by minimality of $m$ and since $C(\text{id}, m)$ excludes $E(\text{id}, m)$), so the **while** loop guard is satisfied and $C(\text{id}, m) \mapsto V(\text{id}, m)$ executes immediately.

If $E(\text{id}, m)$: step $m$'s VERIFY_STEP completes after finite time (the assumption that each event loop iteration finishes in finite time implies every VERIFY_STEP returns, since otherwise **fetch_idle_worker** would eventually block indefinitely). Once the return is processed by some iteration with $\text{id}'$, $E(\text{id}', m) \mapsto C(\text{id}', m)$ occurs, and $C(\text{id}', m) \mapsto V(\text{id}', m)$ follows by the same argument (whether accepted or rejected, step $m$ remains in CV as its minimum-$\text{id}$ element).

Thus $|\text{V}|$ increases by at least 1 after finitely many iterations. After at most $K$ such increases, $|\text{V}| = K + 1$.

$\square$

**Algorithm 2** Pipelined Langevin Speculative Dynamics, ABOBA Variant, No Speculative Error Correction

**Procedure** MAIN():

**input** DraftModel, TargetModelPool, final step count $K$, initial positions and momenta $x_0 = (\mathbf{q}_0, \mathbf{p}_0)$.
  ▷ Initialize step buffers: EP (eligible_pending), CV (conditionally_verified), V (verified)
  EP ← [ ] ;  CV ← [ ] ;  V ← [$(x_0, 0)$]
  ▷ Initialize current draft state $y$, id (count that identifies each draft step over its lifetime)
  $y \leftarrow x_0$ ;  id ← 1
  **while** $|\text{V}| < K + 1$ **do**
    $(y', \tilde{\mathbf{F}}) \leftarrow$ DRAFT_STEP($y$) ▷ Propose next draft state $y'$ and draft forces $\tilde{\mathbf{F}}$ for its verification
    TargetModel ← **fetch_idle_worker** (TargetModelPool) ▷ Blocks as long as all are busy
    **submit** VERIFY_STEP($y$, $y'$, $\tilde{\mathbf{F}}$, id) to TargetModel ▷ Non-blocking
    EP ← EP + [$(y', \text{id})$] ;  id ← id + 1 ;  $y \leftarrow y'$
    (EP, CV, V, $y$) ← PROCESS_NEW_RETURNS(TargetModelPool, EP, CV, V, $y$)
  **end while**
**output** V

---

**Procedure** DRAFT_STEP($y$):                                      [synchronous]

▷ Implements an ABOBA integrator step

  $(\tilde{\mathbf{q}}, \tilde{\mathbf{p}}) \leftarrow y$
  $\tilde{\mathbf{q}} \leftarrow \tilde{\mathbf{q}} + \frac{\Delta t}{2} \mathbf{M}^{-1} \tilde{\mathbf{p}}$
  $\tilde{\mathbf{F}} \leftarrow$ DraftModel.get_forces($\tilde{\mathbf{q}}$)
  $\langle \tilde{\mathbf{p}} \rangle \leftarrow$ [Eq. (8)]($\tilde{\mathbf{F}}$) ;  $\boldsymbol{\Sigma} \leftarrow$ [Eq. (8)]($T, \gamma, \Delta t$)
  $\tilde{\mathbf{p}} \leftarrow \mathcal{N}(\cdot ; \langle \tilde{\mathbf{p}} \rangle, \boldsymbol{\Sigma})$
  $\tilde{\mathbf{q}} \leftarrow \tilde{\mathbf{q}} + \frac{\Delta t}{2} \mathbf{M}^{-1} \tilde{\mathbf{p}}$
  $y' \leftarrow (\tilde{\mathbf{q}}, \tilde{\mathbf{p}})$
  **return** $(y', \tilde{\mathbf{F}})$

**Procedure** VERIFY_STEP($y, y', \tilde{\mathbf{F}}, \text{id}'$):                          [asynchronous]

▷ Runs asynchronously to main process in parallel TARGETMODEL processes

  $(\tilde{\mathbf{q}}, \tilde{\mathbf{p}}) \leftarrow y$;  $(\tilde{\mathbf{q}}', \tilde{\mathbf{p}}') \leftarrow y'$;  $\langle \tilde{\mathbf{p}} \rangle \leftarrow$ [Eq. (8)]($\tilde{\mathbf{F}}$)
  $\tilde{\mathbf{q}} \leftarrow \tilde{\mathbf{q}} + \frac{\Delta t}{2} \mathbf{M}^{-1} \tilde{\mathbf{p}}$ ▷ map $f$ in Thm. 3.1
  $\mathbf{F} \leftarrow$ TargetModel.get_forces($\tilde{\mathbf{q}}$) ;  $\langle \mathbf{p} \rangle \leftarrow$ [Eq. (8)]($\mathbf{F}$) ;  $\boldsymbol{\Sigma} \leftarrow$ [Eq. (8)]($T, \gamma, \Delta t$)
  ▷ **Note**: reflection coupling Eq. (9) acts on $\tilde{\mathbf{p}}'$, i.e. the stored draft momentum **after** sampling, see Thm. 3.1.
  $u \leftarrow$ Uniform([0,1]) ;  $\mathbf{p} \leftarrow$ REFLECTIONVERIFY($\tilde{\mathbf{p}}' ; u \mid \langle \mathbf{p} \rangle, \langle \tilde{\mathbf{p}} \rangle$)
  $\mathbf{q} \leftarrow \tilde{\mathbf{q}} + \frac{\Delta t}{2} \mathbf{M}^{-1} \mathbf{p}$ ▷ map $g$ in Thm. 3.1
  $x' \leftarrow (\mathbf{q}, \mathbf{p})$ ▷ By Thm. 3.1, $x' \sim P(\cdot \mid y)$ if $y' \sim Q(\cdot \mid y)$
  **return** $(y', x', \text{id}')$ to main process

**Procedure** PROCESS_NEW_RETURNS(TargetModelPool, EP, CV, V, $y$)           [synchronous]

  R ← **gather_new_returns**(TargetModelPool) ▷ Returns empty if no new returns
  **for** each $(y', x', \text{id}')$ in R **do**
    **if** $(y', \text{id}') \in$ EP **then** EP ← [$(y'', \text{id}'') \in$ EP $\mid \text{id}'' \neq \text{id}'$] **else continue** ▷ ignore step if flushed
    CV ← CV + [$(x', \text{id}')$]
    **if** $x' \neq y'$ **then** {▷ Flush all downstream steps and roll back draft if the step was rejected}
      EP ← [$(y'', \text{id}'') \in$ EP $\mid \text{id}'' \leq \text{id}'$] ;  CV ← [$(x'', \text{id}'') \in$ CV $\mid \text{id}'' \leq \text{id}'$] ;  $y \leftarrow x'$
    **end if**
    **while** CV $\neq$ [ ] $\wedge$ min(CV[:,1]) < min(EP[:,1] + [$\infty$]) **do**
      V ← V + [CV.**pop**(min(CV[:,1]))]
    **end while**
  **end for**
  **return** (EP, CV, V, $y$)

### D.3. Quantifying the efficiency gain from pipelining.

The pipelined (asynchronous) setup provides a meaningful advantage over the synchronous approach of Alg. 1, which we can make precise. Under identical use of Assump. 1 (i.i.d. Bernoulli rejections at constant rate $\langle \beta \rangle = 1 - \langle \alpha \rangle$), the synchronous speedup $F_{\text{sync}}$ is derived as Theorem 3.8 in Leviathan et al. (2023). Adapted to our notation with time-averaged acceptance rate $\langle \alpha \rangle$ and cost fraction $c$, it reads

$$F_{\text{sync}} = \frac{1 - \langle \alpha \rangle^{L+1}}{(1 - \langle \alpha \rangle)(Lc + 1)}, \tag{22}$$

where $L$ is the lookahead (number of draft steps per synchronous round). Our pipelined speedup formula Eq. (11) (with $\langle \beta \rangle = 1 - \langle \alpha \rangle$) has the simpler form

$$F_{\text{async}} = \frac{1}{1 - \langle \alpha \rangle + c}. \tag{23}$$

A key qualitative difference is that $F_{\text{sync}}$ can never tightly reach the bare draft-target speedup $\frac{1}{c}$, even for perfect acceptance rates $\langle \alpha \rangle \to 1$. This is expected, as the synchronous pipeline can only run at draft throughput for bursts of $L$ steps, after which the draft model must pause for one synchronous verification call. Meanwhile, pipelined LSD can indeed sustain full draft throughput (i.e., speedup $\frac{1}{c}$) when $\langle \alpha \rangle \to 1$. Formally, in the nontrivial case $c < 1$,

$$F_{\text{sync}} = \frac{\sum_{k=0}^{L} \langle \alpha \rangle^k}{Lc + 1} \leq \frac{L + 1}{Lc + 1} < \frac{1}{c} = \lim_{\langle \alpha \rangle \to 1} F_{\text{async}}, \tag{24}$$

where we used the truncated geometric series expansion, followed by $\langle \alpha \rangle \leq 1$, followed by the strict inequality $c < 1$.

## E. Speculative Error Correction Details

Alg. 3 documents our complete implementation of speculative error correction (EC, additions to Alg. 2 highlighted in blue). We obtain the last-known historical error by setting the error cache $\Delta F$ to the force error between the target and draft model on the last step that has been added to the `verified` queue, i.e., committed to the simulation. In initial testing of EC, we used the error relative to the *EC-corrected* draft prediction, however we found that this often leads to a positive self-feedback causing instability. We solve this issue by instead computing the errors relative to the *EC-uncorrected* draft prediction. After this change, we have no longer observed any instability-driven EC failure in our experiments.

Note that the correctness and halting result Thm. D.2 proven for Alg. 2 in Sec. D.2 remains equally valid for Alg. 3: EC changes only the forces that are used to propose a draft step, whereas the target forces are not modified. Therefore, the coupling property from Thm. 3.1, used in Lemma D.4 to prove Thm. D.2, remains valid as the target sampling guarantee is independent of the chosen draft distribution.

## F. Generalization to the (OBABO) Integrator

While the main text develops LSD for the (ABOBA) integrator, which achieves high accuracy for configurational (position-based) averages (Leimkuhler & Matthews, 2013), the closely related (OBABO) splitting is preferable when momentum-based observables such as velocity autocorrelation functions are of primary interest (Leimkuhler & Matthews, 2013). We show here that LSD generalizes to (OBABO) via a time-staggering argument that reduces the problem to the (ABOBA) case.

**The (OBABO) scheme.** With the same notation as Eq. (2), the (OBABO) integrator reads

$$
\begin{aligned}
&\text{(O)} \quad \mathbf{p} \leftarrow e^{-\gamma \Delta t/2} \mathbf{p} + \mathbf{\Sigma}_{1/2}^{\frac{1}{2}} \boldsymbol{\xi} \\
&\text{(B)} \quad \mathbf{p} \leftarrow \mathbf{p} + \Delta t/2 \, \mathbf{F}(\mathbf{q}) \\
&\text{(A)} \quad \mathbf{q} \leftarrow \mathbf{q} + \Delta t \, \mathbf{M}^{-1} \mathbf{p} \\
&\text{(B)} \quad \mathbf{p} \leftarrow \mathbf{p} + \Delta t/2 \, \mathbf{F}(\mathbf{q}) \\
&\text{(O)} \quad \mathbf{p} \leftarrow e^{-\gamma \Delta t/2} \mathbf{p} + \mathbf{\Sigma}_{1/2}^{\frac{1}{2}} \boldsymbol{\xi}',
\end{aligned}
\tag{25}
$$

where $\boldsymbol{\xi}, \boldsymbol{\xi}' \sim \mathcal{N}(\mathbf{0}, \mathbb{I})$ are independent and $\mathbf{\Sigma}_{1/2} := \mathbf{M} k_B T \left(1 - e^{-\gamma \Delta t}\right)$ is the covariance of a half-step (O) update. Note that (A) uses a full time step $\Delta t$ (vs. $\Delta t/2$ in ABOBA), while the (B) and (O) sub-step durations match those of the corresponding (ABOBA) sub-steps.

---

**Algorithm 3** Pipelined Langevin Speculative Dynamics, ABOBA Variant, with Speculative Error Correction

---

**Procedure** MAIN():

**input** `DraftModel`, `TargetModelPool`, final step count $K$, initial positions and momenta $x_0 = (\mathbf{q}_0, \mathbf{p}_0)$.
  ▷ Initialize step buffers: EP (`eligible_pending`), CV (`conditionally_verified`), V (`verified`)
  $\text{EP} \leftarrow [\,]$ ; $\text{CV} \leftarrow [\,]$ ; $\text{V} \leftarrow [(x_0, 0)]$
  ▷ Initialize current draft state $y$, id, error cache $\Delta\mathbf{F}$ for speculative error correction (EC)
  $y \leftarrow x_0$ ; $\text{id} \leftarrow 1$ ; $\Delta\mathbf{F} \leftarrow 0$
  **while** $|\text{V}| < K + 1$ **do**
    $(y', \tilde{\mathbf{F}}) \leftarrow \text{DRAFT\_STEP}(y, \Delta\mathbf{F})$ ▷ Propose next draft state $y'$ and draft forces $\tilde{\mathbf{F}}$ for its verification
    `TargetModel` $\leftarrow$ **fetch_idle_worker** (`TargetModelPool`) ▷ Blocks as long as all are busy
    **submit** VERIFY\_STEP$(y, y', \tilde{\mathbf{F}}, \Delta\mathbf{F}, \text{id})$ to `TargetModel` ▷ Non-blocking
    $\text{EP} \leftarrow \text{EP} + [(y', \text{id})]$ ; $\text{id} \leftarrow \text{id} + 1$ ; $y \leftarrow y'$
    $(\text{EP}, \text{CV}, \text{V}, y, \Delta\mathbf{F}) \leftarrow \text{PROCESS\_NEW\_RETURNS}(\text{TargetModelPool}, \text{EP}, \text{CV}, \text{V}, y, \Delta\mathbf{F})$
  **end while**
**output** V

**Procedure** DRAFT\_STEP$(y, \Delta\mathbf{F})$:         [synchronous]

  $(\tilde{\mathbf{q}}, \tilde{\mathbf{p}}) \leftarrow y$
  $\tilde{\mathbf{q}} \leftarrow \tilde{\mathbf{q}} + \frac{\Delta t}{2} \mathbf{M}^{-1} \tilde{\mathbf{p}}$
  $\tilde{\mathbf{F}} \leftarrow \text{DraftModel.get\_forces}(\tilde{\mathbf{q}})$
  ▷ EC: correct the bare draft prediction by the last-known historical target-draft error
  $\langle\tilde{\mathbf{p}}\rangle \leftarrow [\text{Eq. (8)}](\tilde{\mathbf{F}} + \Delta\mathbf{F})$ ; $\mathbf{\Sigma} \leftarrow [\text{Eq. (8)}](T, \gamma, \Delta t)$
  $\tilde{\mathbf{p}} \leftarrow \mathcal{N}(\cdot\,; \langle\tilde{\mathbf{p}}\rangle, \mathbf{\Sigma})$
  $\tilde{\mathbf{q}} \leftarrow \tilde{\mathbf{q}} + \frac{\Delta t}{2} \mathbf{M}^{-1} \tilde{\mathbf{p}}$
  $y' \leftarrow (\tilde{\mathbf{q}}, \tilde{\mathbf{p}})$
  **return** $(y', \tilde{\mathbf{F}})$

**Procedure** VERIFY\_STEP$(y, y', \tilde{\mathbf{F}}, \Delta\mathbf{F}, \text{id}')$:         [asynchronous]

▷ Runs asynchronously to main process in parallel TARGETMODEL processes
  $(\tilde{\mathbf{q}}, \tilde{\mathbf{p}}) \leftarrow y$; $(\tilde{\mathbf{q}}', \tilde{\mathbf{p}}') \leftarrow y'$; $\langle\tilde{\mathbf{p}}\rangle \leftarrow [\text{Eq. (8)}](\tilde{\mathbf{F}} + \Delta\mathbf{F})$
  $\tilde{\mathbf{q}} \leftarrow \tilde{\mathbf{q}} + \frac{\Delta t}{2} \mathbf{M}^{-1} \tilde{\mathbf{p}}$ ▷ map $f$ in Thm. 3.1
  ▷ EC cache $\Delta\mathbf{F}$ will be updated to $\Delta\mathbf{F}'$ once we add this step to V (assuming it is not flushed)
  ▷ Use **EC-uncorrected** draft forces for the new EC cache $\Delta\mathbf{F}'$, which avoids self-feedback instability
  $\mathbf{F} \leftarrow \text{TargetModel.get\_forces}(\tilde{\mathbf{q}})$ ; $\Delta\mathbf{F}' \leftarrow \mathbf{F} - \tilde{\mathbf{F}}$ ; $\langle\mathbf{p}\rangle \leftarrow [\text{Eq. (8)}](\mathbf{F})$ ; $\mathbf{\Sigma} \leftarrow [\text{Eq. (8)}](T, \gamma, \Delta t)$
  ▷ **Note**: reflection coupling Eq. (9) acts on $\tilde{\mathbf{p}}'$, i.e. the stored draft momentum **after** sampling, see Thm. 3.1.
  $u \leftarrow \text{Uniform}([0, 1])$ ; $\mathbf{p} \leftarrow \text{REFLECTIONVERIFY}(\tilde{\mathbf{p}}'\,; u \mid \langle\mathbf{p}\rangle, \langle\tilde{\mathbf{p}}\rangle)$
  $\mathbf{q} \leftarrow \tilde{\mathbf{q}} + \frac{\Delta t}{2} \mathbf{M}^{-1} \mathbf{p}$ ▷ map $g$ in Thm. 3.1
  $x' \leftarrow (\mathbf{q}, \mathbf{p})$ ▷ By Thm. 3.1, $x' \sim P(\cdot \mid y)$ if $y' \sim Q(\cdot \mid y)$
  **return** $(y', x', \Delta\mathbf{F}', \text{id}')$ to main process

**Procedure** PROCESS\_NEW\_RETURNS$(\text{TargetModelPool}, \text{EP}, \text{CV}, \text{V}, y, \Delta\mathbf{F})$         [synchronous]

  $\text{R} \leftarrow$ **gather_new_returns**(`TargetModelPool`) ▷ Returns empty if no new returns
  **for** each $(y', x', \Delta\mathbf{F}', \text{id}')$ in R **do**
    **if** $(y', \text{id}') \in \text{EP}$ **then** $\text{EP} \leftarrow [(y'', \text{id}'') \in \text{EP} \mid \text{id}'' \neq \text{id}']$ **else continue** ▷ ignore step if flushed
    $\text{CV} \leftarrow \text{CV} + [(x', \Delta\mathbf{F}', \text{id}')]$
    **if** $x' \neq y'$ **then** {▷ Flush all downstream steps and roll back draft if the step was rejected}
      $\text{EP} \leftarrow [(y'', \text{id}'') \in \text{EP} \mid \text{id}'' \leq \text{id}']$ ; $\text{CV} \leftarrow [(x'', \Delta\mathbf{F}'', \text{id}'') \in \text{CV} \mid \text{id}'' \leq \text{id}']$ ; $y \leftarrow x'$
    **end if**
    **while** $\text{CV} \neq [\,] \,\wedge\, \min(\text{CV}[:, 2]) < \min(\text{EP}[:, 1] + [\infty])$ **do**
      $(x', \Delta\mathbf{F}, \text{id}') \leftarrow \text{CV}.\textbf{pop}(\min(\text{CV}[:, 2]))$ ▷ Updates EC cache $\Delta\mathbf{F}$ to the $\Delta\mathbf{F}'$ stored in this step
      $\text{V} \leftarrow \text{V} + [(x', \text{id}')]$
    **end while**
  **end for**
  **return** $(\text{EP}, \text{CV}, \text{V}, y, \Delta\mathbf{F})$

---

**Time-staggering reduces (OBABO) to (ABOBA).** Concatenating successive (OBABO) steps and regrouping yields the pattern

$$\underbrace{\text{OB}}_{\text{init.}} \ \text{A} \ \underbrace{\text{BOOB}}_{=\,\text{BOB}} \ \text{A} \ \underbrace{\text{BOOB}}_{=\,\text{BOB}} \ \text{A} \cdots \tag{26}$$

Each (BOOB) block merges into a (BOB) step of the form Eq. (8): the two adjacent $O(\Delta t/2)$ half-steps compose to a single $O(\Delta t)$ full step, since the friction factors multiply as $e^{-\gamma \Delta t/2} \cdot e^{-\gamma \Delta t/2} = e^{-\gamma \Delta t}$ and the noise covariances sum as $\boldsymbol{\Sigma}_{1/2}(1 + e^{-\gamma \Delta t}) = \boldsymbol{\Sigma}$. Moreover, both (B) half-steps within each (BOOB) block evaluate forces at the same position—set by the preceding (A) step, which is the only sub-step that modifies $\mathbf{q}$.

Define *staggered states* $\bar{y}_n$ at the junctions between (BOB) and (A) blocks in Eq. (26), offset from the (OBABO) outputs by an initial (OB) half-step: $\bar{y}_0 := \text{OB}(y_0)$. The staggered transition $\bar{y}_{n-1} \mapsto \bar{y}_n$ reads

$$\bar{y}_n = \text{BOB}\big(\text{A}(\bar{y}_{n-1})\big),$$

which has the structure of Thm. 3.1 with $f = \text{A}$ (the deterministic, force-independent position update) as the pre-processing map, the (BOB) Gaussian momentum update as the force-dependent stochastic part $P'$, $Q'$ (coupled identically to the (ABOBA) case via REFLECTIONVERIFY) and $g = \text{id}$ as the (trivially invertible) post-processing map. Thm. 3.1 therefore guarantees both the coupling property and the maximal acceptance rate for LSD on the staggered states.

**Recovery of (OBABO) output states.** It remains to extract the actual (OBABO) output states $y_n$ from the verified staggered trajectory. The (OBABO) output $y_n$ sits at the midpoint of the merged $O(\Delta t)$ step within the (BOB) block that maps $\bar{y}_{n-1} \mapsto \bar{y}_n$, i.e., after the first (BO) half of the corresponding (BOOB) but before the second (OB) half.

Write $\bar{\mathbf{p}}_{n-1}, \bar{\mathbf{p}}_n$ for the momenta of the staggered states $\bar{y}_{n-1}, \bar{y}_n$ and $\mathbf{q}_n$ for their shared position (the (BOB) block leaves positions fixed at the value set by the preceding (A) step). Both (B) half-steps inside the block evaluate the same target force $\mathbf{F}_n := \mathbf{F}(\mathbf{q}_n)$, which is *already known*: it was computed when the staggered transition $\bar{y}_{n-1} \mapsto \bar{y}_n$ was verified through REFLECTIONVERIFY, so no additional force evaluation is needed to recover $y_n$. Undoing the surrounding (B) half-steps recovers the momenta entering and leaving the merged $O(\Delta t)$ step,

$$\mathbf{p}_{\text{in}} = \bar{\mathbf{p}}_{n-1} + \tfrac{\Delta t}{2}\, \mathbf{F}_n, \qquad \mathbf{p}_{\text{out}} = \bar{\mathbf{p}}_n - \tfrac{\Delta t}{2}\, \mathbf{F}_n, \tag{27}$$

so both carry a force-dependent shift from the (B) steps. The merged $O(\Delta t)$ step itself decomposes into two $O(\Delta t/2)$ half-steps driven by independent noise $\boldsymbol{\xi}_1, \boldsymbol{\xi}_2 \sim \mathcal{N}(\mathbf{0}, \mathbb{I})$:

$$\mathbf{p}_{\text{mid}} = e^{-\gamma \Delta t/2}\, \mathbf{p}_{\text{in}} + \boldsymbol{\Sigma}_{1/2}^{\frac{1}{2}}\, \boldsymbol{\xi}_1,$$
$$\mathbf{p}_{\text{out}} = e^{-\gamma \Delta t/2}\, \mathbf{p}_{\text{mid}} + \boldsymbol{\Sigma}_{1/2}^{\frac{1}{2}}\, \boldsymbol{\xi}_2, \tag{28}$$

and the (OBABO) output momentum is the intermediate $\mathbf{p}_{\text{mid}}$. Through Eq. (27), $\mathbf{p}_{\text{mid}}$ has the *force-dependent* mean $e^{-\gamma \Delta t/2}\, \mathbf{p}_{\text{in}} = e^{-\gamma \Delta t/2}\big(\bar{\mathbf{p}}_{n-1} + \tfrac{\Delta t}{2}\, \mathbf{F}_n\big)$; it is *not* a centered Gaussian.

The staggered LSD step fixes $\bar{\mathbf{p}}_n$ and hence, via Eq. (27), both $\mathbf{p}_{\text{in}}$ and $\mathbf{p}_{\text{out}}$ together with the weighted noise sum

$$\mathbf{w} := \boldsymbol{\Sigma}_{1/2}^{-\frac{1}{2}}\big(\mathbf{p}_{\text{out}} - e^{-\gamma \Delta t}\, \mathbf{p}_{\text{in}}\big) = \boldsymbol{\Sigma}_{1/2}^{-\frac{1}{2}}\Big(\bar{\mathbf{p}}_n - e^{-\gamma \Delta t}\, \bar{\mathbf{p}}_{n-1} - \big(1 + e^{-\gamma \Delta t}\big)\tfrac{\Delta t}{2}\, \mathbf{F}_n\Big) = e^{-\gamma \Delta t/2}\, \boldsymbol{\xi}_1 + \boldsymbol{\xi}_2, \tag{29}$$

which does not fix $\boldsymbol{\xi}_1, \boldsymbol{\xi}_2$ individually. Note that $\mathbf{w}$ depends on the (B)-step force $\mathbf{F}_n$ through $\mathbf{p}_{\text{in}}$ and $\mathbf{p}_{\text{out}}$. Since $\boldsymbol{\Sigma}_{1/2}$ is diagonal and $\boldsymbol{\xi}_1, \boldsymbol{\xi}_2$ are independent standard Gaussians, the conditional distribution of $\boldsymbol{\xi}_1$ given $\mathbf{w}$ factorizes across components:

$$\boldsymbol{\xi}_1 \mid \mathbf{w} \ \sim \ \mathcal{N}\left(\frac{e^{-\gamma \Delta t/2}}{1 + e^{-\gamma \Delta t}}\, \mathbf{w}, \ \frac{1}{1 + e^{-\gamma \Delta t}}\, \mathbb{I}\right). \tag{30}$$

Drawing $\boldsymbol{\xi}_1^*$ from Eq. (30) and substituting into the first line of Eq. (28) yields the (OBABO) output momentum

$$\mathbf{p}_{\text{mid}} = e^{-\gamma \Delta t/2}\big(\bar{\mathbf{p}}_{n-1} + \tfrac{\Delta t}{2}\, \mathbf{F}_n\big) + \boldsymbol{\Sigma}_{1/2}^{\frac{1}{2}}\, \boldsymbol{\xi}_1^*, \tag{31}$$

which makes the contribution of the (B)-step force explicit. As (BOB) leaves positions unchanged, the output position $\mathbf{q}_n$ coincides with that of the staggered state $\bar{y}_n$. In summary, one can run LSD entirely in the staggered frame and recover (OBABO) trajectory outputs by sampling a half-step into the merged Ornstein–Uhlenbeck step, conditioned on the staggered result and shifted by the (B)-step forces $\mathbf{F}_n$. Because these target forces were already evaluated during verification, the recovery reuses them and requires *no* recomputation of $\mathbf{F}$.

## G. Kernel parity test of LSD and target trajectory distribution

**Background.** Maximum Mean Discrepancy (MMD) (Gretton et al., 2012) is a distance measure between probability distributions based on their embeddings in a Reproducing Kernel Hilbert Space (RKHS). Given a kernel $k$ with associated RKHS $\mathcal{H}$, a distribution $P$ can be represented by its **mean embedding**:

$$\mu_P = \mathbb{E}_{X \sim P}[k(\cdot, X)] \in \mathcal{H}.$$

The MMD between two distributions $P$ and $Q$ is then defined as the RKHS norm of the difference between their embeddings:

$$\text{MMD}(P, Q) = \|\mu_P - \mu_Q\|_{\mathcal{H}}.$$

Expanding the squared norm using the reproducing property yields:

$$\text{MMD}^2(P, Q) = \mathbb{E}_{X, X'}[k(X, X')] - 2\mathbb{E}_{X, Y}[k(X, Y)] + \mathbb{E}_{Y, Y'}[k(Y, Y')].$$

Given samples $\{x_i\}_{i=1}^n \sim P$ and $\{y_j\}_{j=1}^m \sim Q$, the **biased empirical estimator** replaces expectations with sample averages (including diagonal terms):

$$\widehat{\text{MMD}}_b^2 = \frac{1}{n^2} \sum_{i,j} k(x_i, x_j) - \frac{2}{nm} \sum_{i,j} k(x_i, y_j) + \frac{1}{m^2} \sum_{i,j} k(y_i, y_j).$$

This estimator is biased because including terms like $k(x_i, x_i)$ introduces positive bias, though it remains consistent. We use it instead of the unbiased estimator (which corrects for the diagonal contribution) to avoid MMD negative values. A natural question is whether $\text{MMD}(P, Q) = 0$ implies that $P$ and $Q$ are the same distribution. This property holds when the kernel $k$ is **characteristic**, meaning that the mean embedding map $P \mapsto \mu_P$ is injective. Under this condition, $\text{MMD}(P, Q) = 0$ if and only if $P = Q$ in the sense of **equality in distribution**—that is, $P(A) = Q(A)$ for all measurable sets $A$, or equivalently, the two distributions assign the same probability to every event. Many commonly used kernels are characteristic, including the Gaussian (RBF) kernel $k(x, y) = \exp(-\|x - y\|^2 / 2\sigma^2)$ and the Laplacian kernel. This property makes MMD a powerful tool for two-sample testing, as it provides a true metric on the space of probability distributions when a characteristic kernel is used.

**Experimental setup.** We sample 1000 length-10 trajectories of for 108-atom FCC copper, repeating this with 3 random seeds for each model (UMA-S, ORB-V3-DIRECT-20-OMAT and their LSD combination). Every sampled trajectory is therefore a vector in $\mathbb{R}^{6 \times 108 \times 10}$ (positions and momenta of 108 atoms $(\mathbf{x}_n, \mathbf{p}_n) \in \mathbb{R}^{6 \times 108}$, concatenated for 10 time steps $n$). We then compute the maximum mean discrepancy (MMD) between the pairs of high-dimensional trajectory point clouds, using the Gaussian kernel with bandwidth $\sigma$ set to the median $L^2$ distance between vectors. To obtain Fig. 8, we repeat this procedure for all 6 pairs of different seeds for the same pair of models, and take the average and empirical standard deviation over the obtained MMD values.

**Further MMD results.** We provide additional MMD results for the combination of a UMA-S target and an EMT on the same 108-atom copper system in Fig. 9.

## H. Relaxation of Target Model Guarantees

**Force slack.** If EC or other alignment is insufficient, a plausible final option is to controllably relax the guaranty of matching the exact target trajectory distribution. The idea of *force slack* is to accept a small component-wise force error (slack) $\epsilon_F > 0$ and, at verification time, push the target forces as much towards the draft forces as admissible within an $L_\infty$ ball of norm $\epsilon_F$. In other words, we substitute the true target forces $\mathbf{F}$ by $\mathbf{F}_{(\epsilon)} = \mathbf{F} + \text{clip}(-\epsilon_F; \tilde{\mathbf{F}} - \mathbf{F}; \epsilon_F)$, where the function $\text{clip}(l, \mathbf{x}, u) \colon \mathbb{R}^{3N} \to \mathbb{R}^{3N}$, $x^{(i)} \mapsto \min\left(\max\left(x^{(i)}, l\right), u\right)$, with $l < u \in \mathbb{R}$, acts per vector component $1 \le i \le 3N$. Intuitively, this has the effect of pushing the two Gaussians of the momentum step (cf. Figs. 2a to 2c) closer together. Unlike speculative error correction, which modifies only the drafting process, force slack shifts the target forces at verification time and therefore breaks the exact coupling to the target distribution. Instead, LSD with slack couples the draft distribution to the distribution of the relaxed target field $\mathbf{F}_{(\epsilon)}$ with $\|\mathbf{F}_{(\epsilon)}(\mathbf{q}) - \mathbf{F}(\mathbf{q})\|_\infty \le \epsilon_F \; \forall \mathbf{q} \in \mathbb{R}^{3N}$.

**Negative impact on diffusion coefficients.** Unfortunately, we find empirically that the usage of force slack significantly degrades the estimation of diffusion coefficients, cf. Fig. 10. We therefore recommend using LSD with its strict guarantees.

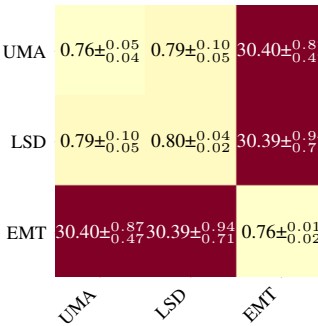

*Figure 9.* MMD distances (Gaussian kernel, median distance $\sigma$, amplified by $10^3$) for UMA-S, EMT and their LSD combination on 108-atom copper. LSD and UMA-S are as close to each other as UMA-S to itself given different random seeds, while EMT predicts a significantly different trajectory distribution.

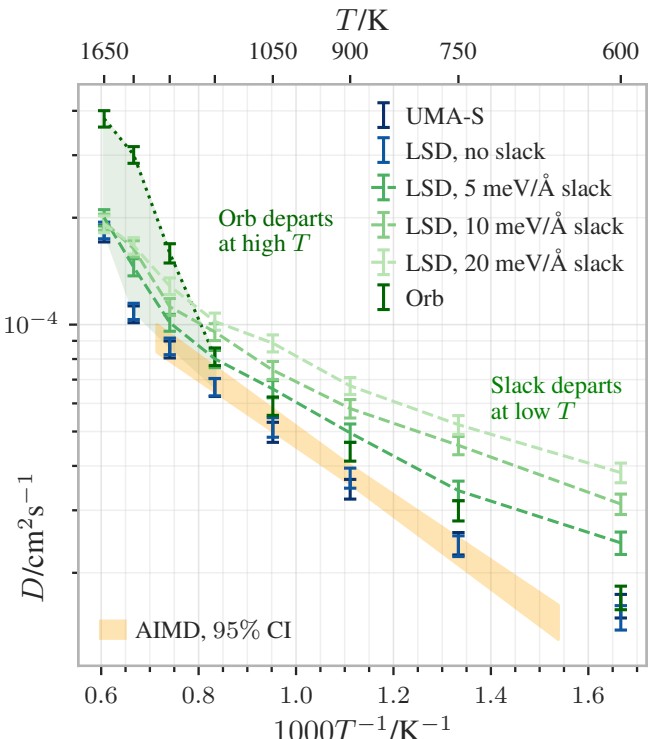

*Figure 10.* Diffusion coefficients for UMA-S, Orb-v3-direct-omat and their LSD combination degrade with increasing slack, with diffusivity being overestimated even more than by pure Orb in the low-temperature range.

## I. Speedups for Bulk Water

We measure further speedups on bulk water. Due to the lower temperature of $300K$, rejection rates are generally significantly higher and speedups at the fixed, low friction (high friction timescale) of $\tau = 1000$fs which we used for copper are comparably low. We therefore plot three different friction timescales to show a wider range of possible results. It becomes evident that the rejection rate and therefore the speedup is highly dependent on the chosen friction range.

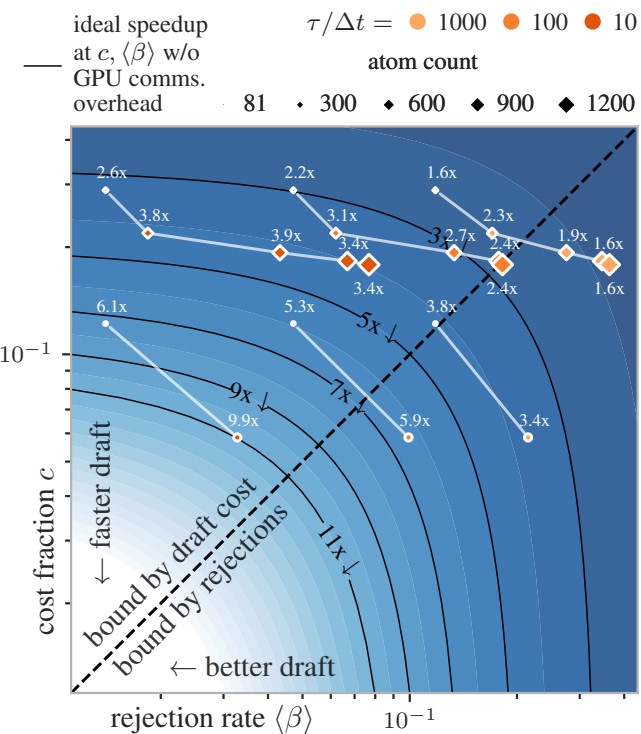

*Figure 11.* Measured LSD walltime speedups (white labels) for (target, draft): (`UMA-S`, `Orb-v3-direct`) (◆) at 8 GPUs and (`UMA-M`, `Orb-v3-direct`) (●) at 32 GPUs, on bulk water at multiple friction timescales $\tau$; plotted against the measured draft / target cost fractions $c$ and avg. rejection rates $\langle\beta\rangle$. All seen speedups are bounded by the ideal Eq. (11), plotted as the contour landscape.

## J. Radial Distribution Functions for Bulk Water

To further confirm that LSD preserves the target model distribution in practice, we computed the O–O and H–H radial distribution functions (RDFs) for bulk water using the same `UMA-S` target and `UMA-tiny-omol` draft setup as studied in the main body. We repeated the comparison at two different Langevin friction values ($\gamma = 0.1\,\mathrm{ps}^{-1}$ and $\gamma = 1\,\mathrm{ps}^{-1}$) to confirm that the result holds across different dynamical regimes.

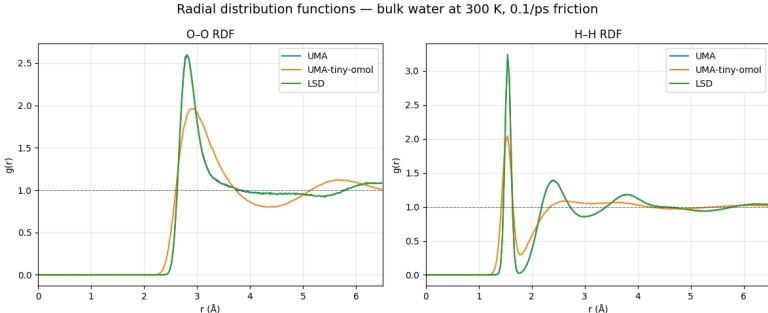

*Figure 12.* O–O and H–H radial distribution functions for bulk water at $\gamma = 0.1\,\mathrm{ps}^{-1}$, comparing the target model (`UMA-S`), LSD (target + draft), and the draft model (`UMA-tiny-omol`). LSD and target curves are indistinguishable, while the draft model deviates significantly.

In both cases, the LSD and target model RDF curves agree to a near-indistinguishable level, while the draft model curve deviates noticeably. This is fully consistent with our strong mathematical guarantees: since LSD samples exactly from the target model trajectory distribution (up to machine precision), all observables computed from LSD trajectories must match those of the target model.

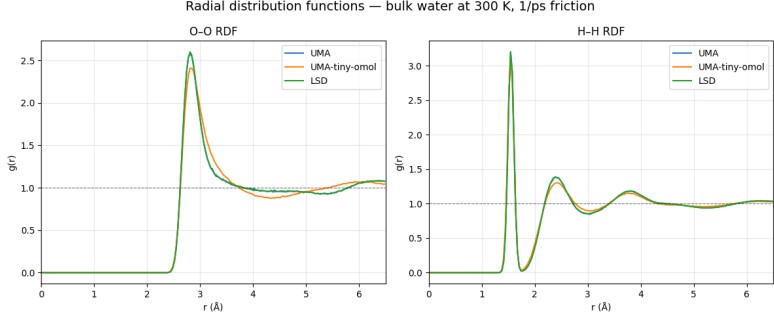

*Figure 13.* O–O and H–H radial distribution functions for bulk water at $\gamma = 1\,\mathrm{ps}^{-1}$. LSD again matches the target model to a near-indistinguishable level, confirming distributional parity across friction regimes.

## K. Rejection Rate Scaling for Alanine Oligopeptides and LGPS

To evaluate how well the rejection rate scaling law (Eq. (12)) generalizes beyond the FCC copper system studied in the main body, we ran new comparisons of observed rejection rates against theory on four alanine oligopeptides (ala2, ala4, ala6, ala8) (300 K, $\gamma = 5\,\mathrm{ps}^{-1}$, see Tabs. 2 and 3) and our LGPS solid electrolyte (1200 K, $\gamma = 0.2\,\mathrm{ps}^{-1}$, see Tabs. 4 and 5). The empirical constant $\varepsilon$ is averaged across its estimates from all atom counts for a more robust fit. Several observations follow from these results.

1. EC reduces rejection rates across the board. For LGPS, EC is essential to access the low-friction regime (in which Li ions can diffuse freely) at practical rejection rates, reducing rejections from $\sim 41\%$ to $\sim 11\%$ at 200 atoms.

2. For the alanine peptides, the scaling fit systematically overestimates rejection rates at low atom counts and underestimates at high atom counts—i.e., rejections grow somewhat faster than the homogeneous scaling law predicts.

This points to a partial breakdown of the spatial independence assumption (Assump. 2), as expected for a heterogeneous system with non-uniform force errors. Nevertheless, the theory models the correct qualitative trend, and the true increase in rejections, while slightly exceeding predictions, is slow enough to define a practical system-size regime for LSD.

*Table 2.* Observed vs. predicted rejection rates for alanine oligopeptides without error correction (300 K, $\gamma = 5\,\mathrm{ps}^{-1}$).

| System | Atom count | Rejection rate [%] | Theory [%] |
|---|---|---|---|
| ala2 | 22 | $4.5 \pm 0.8$ | 5.6 |
| ala4 | 42 | $7.3 \pm 0.5$ | 7.7 |
| ala6 | 62 | $10.4 \pm 0.9$ | 9.3 |
| ala8 | 82 | $12.2 \pm 0.5$ | 10.7 |

*Table 3.* Observed vs. predicted rejection rates for alanine oligopeptides with error correction (300 K, $\gamma = 5\,\mathrm{ps}^{-1}$).

| System | Atom count | Rejection rate [%] | Theory [%] |
|---|---|---|---|
| ala2 | 22 | $3.5 \pm 0.5$ | 4.2 |
| ala4 | 42 | $5.9 \pm 0.4$ | 5.8 |
| ala6 | 62 | $7.6 \pm 1.0$ | 7.1 |
| ala8 | 82 | $8.8 \pm 1.1$ | 8.1 |

*Table 4.* Observed vs. predicted rejection rates for LGPS without error correction (1200 K, $\gamma = 0.2\,\mathrm{ps}^{-1}$).

| Atom count | Rejection rate [%] | Theory [%] |
|---|---|---|
| 200 | $77.7 \pm 1.3$ | 80.8 |
| 400 | $92.8 \pm 1.3$ | 93.5 |
| 600 | $97.9 \pm 0.4$ | 97.6 |
| 800 | $99.5 \pm 0.3$ | 99.1 |

*Table 5.* Observed vs. predicted rejection rates for LGPS with error correction (1200 K, $\gamma = 0.2\,\mathrm{ps}^{-1}$).

| Atom count | Rejection rate [%] | Theory [%] |
|---|---|---|
| 200 | $18.0 \pm 1.0$ | 16.7 |
| 400 | $23.1 \pm 0.9$ | 23.5 |
| 600 | $27.6 \pm 1.2$ | 28.6 |
| 800 | $32.1 \pm 0.8$ | 32.8 |

## L. Extended Graph Parallelism Ceiling Analysis

In Sec. 5 (Figure 6), we compare LSD throughput against graph parallelism (GP) as a function of atom count, observing that LSD outperforms GP by up to an order of magnitude in the $\sim 1000$-atom regime. A natural question is whether the "Graph Parallelism Ceiling" observed in that figure is specific to the UMA model and implementation used, or whether it reflects a general feature of current state-of-the-art MLIPs.

**Minimum latency across models.** To answer this, we benchmarked single-model inference times on a minimal two-atom graph for a variety of modern MLIPs on 80 GB H100 GPUs. These minimum latencies define, for each model, the best possible throughput under *any* multi-GPU implementation based on spatial domain decomposition, since even a one-atom system cannot be evaluated faster. They therefore represent a hard ceiling on GP throughput that is independent of the specific implementation.

*Table 6.* Minimum inference latency per step for a selection of state-of-the-art MLIPs on an H100 GPU (two-atom graph). These define a universal throughput ceiling for any spatial parallelism approach.

| Model | Min. latency (ms) |
|---|---|
| UMA-s (compiled) | 20 |
| MACE-MP (small, cuEQ) | 18 |
| MACE-MP (medium, cuEQ) | 25 |
| MACE-OMoL (large, no cuEQ) | 25 |
| SevenNet Omni FlashTP | 20 |
| ORBv3-conservative | 10 |

Minimum inference times of around 20 ms are a general feature across current SOTA MLIPs, with the sole exception of ORBv3-conservative at $\sim 10$ ms. Most modern highly accurate MLIPs therefore cannot exceed $\sim 50$ steps per second (qps) under any GP implementation, even assuming perfectly linear strong scaling. Note that this paper uses ORBv3-direct as the LSD *draft* model (not conservative), which is faster still; the above table refers to models that would play the role of the target model in LSD.

**LSD throughput at lower GPU counts.** The main body focuses on the theoretical throughput optimum from Sec. 3.2. To address the practical scenario where fewer GPUs are available, we additionally report measured LSD inference speeds at lower GPU counts in Tab. 7. These results confirm that LSD's advantage over GP persists across different hardware configurations in the $\sim 1000$-atom regime.

These results demonstrate that LSD provides meaningful speedups over GP already at 2–4 GPUs for moderate atom counts, and that the advantage grows with additional GPUs. At larger atom counts ($\gtrsim 2000$), rejection rates increase (cf. Eq. (12)) and LSD throughput falls below the GP ceiling, consistent with the crossover observed in the main body.

*Table 7.* Measured LSD inference throughput (queries per second, qps) at selected GPU counts and atom counts for the UMA-S + ORBv3-direct draft-target pair on copper. GP throughput ceiling from Tab. 6 shown for reference.

| GPU count | Atom count | LSD qps | GP ceiling (qps) |
|---|---|---|---|
| 2 | 108 | $\sim 42$ | $\leq 50$ |
| 2 | 500 | $\sim 35$ | $\leq 50$ |
| 4 | 108 | $\sim 71$ | $\leq 50$ |
| 4 | 500 | $\sim 58$ | $\leq 50$ |
| 8 | 108 | $\sim 100$ | $\leq 50$ |
| 8 | 2000 | $\sim 22$ | $\leq 50$ |

## M. Ramachandran Plots for Alanine Dipeptide

Alanine dipeptide is a standard benchmark for evaluating conformational sampling in MD, as its Ramachandran plot (the distribution of backbone dihedral angles $\phi$ and $\psi$) is well characterized experimentally and computationally. We use it here to provide two additional validations: first, that LSD preserves target model distributions for a biologically relevant observable, and second, to discuss the solvation approximations used in our ala2 experiments throughout this work.

All Ramachandran plots in this section were obtained from 15 ns Langevin trajectories at $T = 500$ K (for faster convergence in a single-trajectory setup) and $\gamma = 1\,\mathrm{ps}^{-1}$, with frames recorded every 50 ps.

### M.1. LSD preserves the target Ramachandran distribution

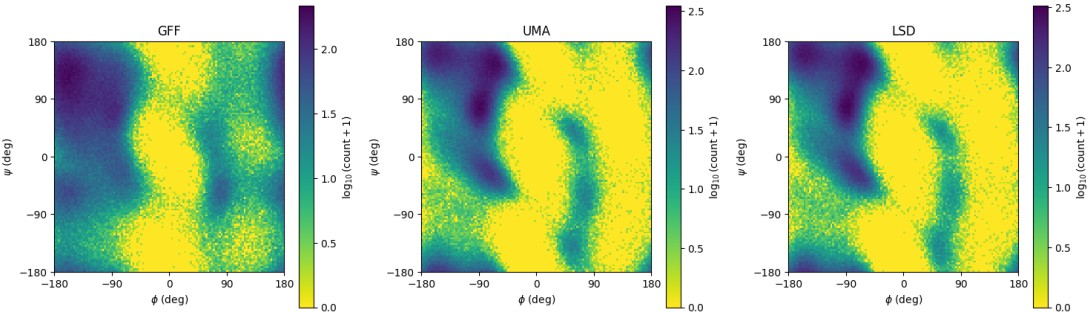

*Figure 14.* Ramachandran plots of ala2 at 500 K with implicit solvent. Left: the Grimme Force Field (GFF) draft model. Middle: the UMA-S target model. Right: LSD using GFF as draft and UMA-S as target. The LSD and UMA-S distributions are visually indistinguishable, while GFF shows a washed-out basin structure and an overpopulation of the $\phi > 0$ half.

Fig. 14 compares the Ramachandran distributions obtained with the classical Grimme Force Field (GFF), as a draft model, UMA-S as the target model, and their LSD combination. The GFF result is qualitatively crude: the basin structure is washed out and the positive-$\phi$ half of the plot is strongly overpopulated relative to UMA-S. In contrast, the UMA-S and LSD Ramachandran plots are essentially indistinguishable. This provides another independent parity result, confirming on a biologically relevant conformational observable that LSD faithfully preserves the target model distribution of trajectories.

### M.2. Solvation approximations

In all alanine dipeptide experiments in this work, we used an implicit solvent model rather than explicit solvation. The reason is that LSD in its current form and our model pairs is not yet practical with explicit solvent, as the presence of many water molecules (which dominate the atom count of the solvated system) drives up the rejection rate to impractical levels.

Fig. 15 compares Ramachandran plots from three solvation approaches using Orb-v3: explicit solvation in 100 water molecules, implicit solvent, and vacuum. The implicit solvent model used here is crude, applying the partial charges, solvent-accessible surface area (SA) terms and other corrections provided by GFF as a fast additive correction to the MLIP energy. While this clearly improves over the vacuum result (which e.g. strongly overpopulates the $C_{7eq}$ basin that is known to be suppressed in aqueous solution) the implicit solvent correction remains incomplete: the $C_{7eq}$ basin is still visibly overpopulated compared to the explicit solvent reference.

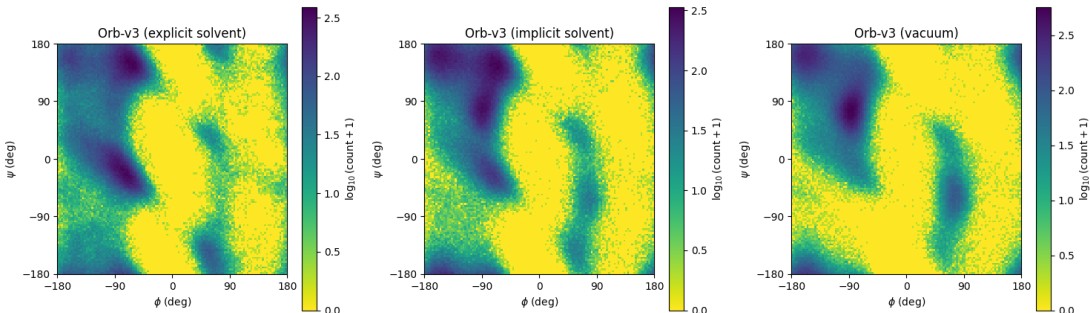

*Figure 15.* Ramachandran plots of ala2 at 500 K under three solvation conditions, all obtained with Orb-v3. Left: explicit solvation in 100 water molecules. Middle: implicit solvent (using GFF-provided partial charges, SA terms, etc. as a correction to the MLIP energy). Right: vacuum (no solvent). The implicit solvent model corrects the vacuum result but still overpopulates basins like $C_{7eq}$ relative to explicit solvation.

For the parity study in this work, this approximation is acceptable, as the implicit solvent is applied consistently to both draft and target and our main claim is distributional equivalence between LSD and the target, not physical accuracy of the solvation model. However, for investigations beyond parity, better implicit solvent approximations tuned specifically to the baseline MLIP and its implicitly learned partial charge model should be used.

Looking forward, it would be worthwhile to investigate how to reduce LSD rejection rates in explicitly solvated systems. One promising direction is to fine-tune the draft model directly on target-force-labeled, explicitly solvated structures (or even on pure bulk solvent) in order to drive down the rejection rates on the solvent atoms that dominate the atom count of the solvated system.

## N. Models and Inference Settings

All experiments in the paper were done on 80GB NVidia H100 GPUs, Python 3.12, Pytorch-2.8. Tab. 8 lists the models that were used and their source.

*Table 8.* Models and settings used in the paper

| Model | Settings | Source/Description |
| --- | --- | --- |
| `UMA-s-1p1` | turbo (TF32, compiled, merged MoLE) | public v1.1 release on Huggingface |
| `UMA-m-1p1` | TF32, activation checkpointing, merged MoLE | public v1.1 release on Huggingface |
| `UMA-tiny-omol` | turbo (TF32, compiled, merged MoLE) | UMA arch with single task, L2M2, 2 Layer, 500k Active params, 18M total params (32 MoLE experts) trained on Omol25 dataset |
| `UMA-tiny-omat` | turbo (TF32, compiled, merged MoLE) | UMA arch with single task, L2M2, 2 Layer, 500k Active params, 18M total params (32 MoLE experts) trained on Omat24 dataset |
| `Orb-v3-direct_20_omat` | default (TF32, compiled) | public v3 release |
| `orb-v3-conservative-inf-omat` | default (TF32, compiled) | public v3 release |
| `orb-v3-conservative-inf-omol` | default (TF32, compiled) | public v3 release |

