# OpenReview forum: "Speculative Sampling For Faster Molecular Dynamics"
_ICML.cc/2026/Conference — ICML 2026 regular_

### Official Review · Reviewer_JcAk · 2026-03-10

**Soundness:** 3
**Presentation:** 3
**Significance:** 2
**Originality:** 3
**Overall Recommendation:** 5
**Confidence:** 4

**Summary:**

This paper introduces a method to accelerate molecular dynamics simulations by adapting speculative decoding from LLMs to underdamped Langevin dynamics. The core idea is to use a fast, approximate draft model (e.g., a lightweight MLIP) to propose simulation steps, which are then verified in parallel by a slower, more accurate target model (e.g., a universal MLIP) using reflection-maximal couplings. The authors provide theoretical guarantees that the distribution of LSD-sampled trajectories matches that of the target model, derive scaling laws for rejection rates and speedups, and introduce practical improvements like speculative error correction. Experiments demonstrate significant speedups (up to 9×) while preserving target statistics.

**Compliance With Llm Reviewing Policy:**

Affirmed.

**Final Justification:**

I've raised my score a point.   Even though I have some lingering concerns about the practicality of the approach, the authors have done enough to convince me that the idea has merit and any good ideas in this field are valuable.

**Key Questions For Authors:**

How does the rejection rate scaling (Eq. 20) hold for disordered systems like proteins or organic electrolytes, where force errors are likely correlated and violate the spatial independence assumption?

Do you have preliminary results on such systems?

Could you provide a more detailed algorithmic description of EC (perhaps as pseudocode) and discuss its stability conditions?

Is there any risk of error accumulation or oscillatory behavior over long trajectories?

Can you sketch how LSD would be applied to a BAOAB integrator?

Would the verification step remain the same (i.e., identifying a "force-dependent block"), or would the coupling need to be fundamentally adapted?

**Limitations:**

While the theoretical framework is sound, the empirical validation is primarily confined to simple metallic systems (FCC copper) with limited exploration of disordered or heterogeneous systems (e.g., proteins, liquids, polymers) where force-field errors may be correlated and violate the spatial independence assumption underlying the rejection rate scaling law. Additionally, although Theorem 3.1 suggests generalizability to other integrators via the "contiguous sampling block" concept, the paper does not explicitly demonstrate how LSD would be applied to widely used schemes like BAOAB or UBU, leaving uncertainty about whether the method requires integrator-specific modifications or extends seamlessly. These limitations suggest avenues for future work, including validation on complex molecular systems and explicit treatment of alternative integrators.

**Strengths And Weaknesses:**

Strengths

Novel and timely contribution: The serial nature of MD simulations is a well-known problem most MD practitioners accept. The paper bridges this critical gap by applying speculative sampling to molecular dynamics, addressing a real bottleneck in MLIP-driven simulations. The adaptation to underdamped Langevin dynamics is technically sound and using splitting integrators demonstrates domain expertise.
Solid theoretical background: The reduction of MD step verification to a reflection-maximal coupling on the Gaussian momentum update (BOB step) is elegant. Theorem 3.1 (Pre-/post-processing) is correctly proven and provides a clean framework for extending the method to various integrators. The derivation of the rejection rate scaling law (Eq. 20) is insightful and grounded in testable assumptions.

Clear exposition and visualizations: The paper is well-written and accessible.

Figures (especially the visual explanation of REFLECTIONVERIFY in Figure 2 and the Pareto analysis in Figure 4) are effective at communicating complex ideas.

Comprehensive evaluation of guarantees: The experiments in Section 5.3 (non-conservative heating, Li diffusion, MMD trajectory tests) convincingly demonstrate that LSD preserves the target distribution, which is the primary theoretical promise of the method.
Practical impact: The speedups (up to 9×) are meaningful, and the analysis of trade-offs between cost fraction and rejection rate (Figure 4) provides practical guidance for practitioners choosing draft-target pairs. The speculative error correction (EC) technique shows promising results in reducing rejection rates.

Weaknesses

Limited generalizability of empirical validation: The experimental evaluation relies heavily on FCC copper, a simple, ordered metallic system. While this is a sensible starting point, it does not represent the diversity of systems MD is used for (e.g., disordered proteins, liquid electrolytes, polymers, interfaces). The rejection rate scaling law (Eq. 20) assumes spatial homogeneity and independence of force errors (Assumption 2), which may break down in complex systems with long-range interactions. The lone example of bulk water (Table 1, Appendix G) is insufficient to establish broad applicability.

Integrator choice and generalizability: The method is presented using the ABOBA splitting scheme for simplicity, where the "BOB" block cleanly isolates momentum updates. However, modern MD practice often favors BAOAB for configurational sampling. While Theorem 3.1 suggests generalization is possible via the "contiguous sampling block" concept, the paper does not explicitly demonstrate how LSD would be applied to BAOAB or UBU integrators. This leaves uncertainty about the method's flexibility and whether the same verification strategy applies without modification.

Speculative Error Correction (EC) is underspecified: EC reduces rejection rates by up to 75% in Figure 3, yet Section 3.3 provides only a high-level description. Key details are missing:
    How is the "most recent historical error" cached and applied across steps?
    What are the failure modes or stability concerns when the error changes sign rapidly or exhibits oscillatory behavior?
    Is there an ablation study isolating the effect of EC from other components?
    Can EC lead to error accumulation over long trajectories?

This lack of detail makes it difficult to fully trust the dramatic improvement.

Comparison to Graph Parallelism reveals a limitation: The crossover where graph parallelism (GP) outperforms LSD at larger atom counts (Figure 5) is presented as a neutral observation, but it actually highlights a fundamental algorithmic bottleneck: rejection rates increase with system size (Eq. 20), capping LSD's scalability. The "fixed compute overhead" cited for GP may be implementation-specific and potentially optimizable, whereas the rejection-rate scaling is inherent to LSD. The paper should frame this trade-off more transparently as a size-dependent limitation rather than a crossover between comparable methods.

Missing citation: The connection to diffusion models is appropriate, though the authors could cite Critically Damped Langevin Diffusion Models by Tim Dockhorn et al for completeness.

Specific Justification for Significance Score: While the core idea is novel and technically sound, the demonstrated practical impact is limited. The method is developed and validated primarily for the ABOBA splitting scheme, whereas BAOAB is widely preferred in modern Langevin MD for its superior configurational sampling properties. The paper does not explicitly demonstrate compatibility with BAOAB or other commonly used integrators, leaving uncertainty about adoption in realistic workflows. Additionally, empirical validation is concentrated on homogeneous systems (primarily FCC copper), while the rejection-rate scaling assumptions rely on spatial independence and homogeneity of force errors that may not hold in heterogeneous systems such as proteins or complex electrolytes. Since LSD’s speedup is fundamentally capped by rejection rates, deviations from these assumptions could significantly limit scalability. As a result, the demonstrated scope of impact is narrower than suggested, which motivates a moderate significance score.

---

> ### Author Rebuttal · Authors · 2026-03-31
>
> We thank you for your thorough feedback.
>
> > How does the rejection rate scaling (Eq. 20) hold for disordered systems like proteins or organic electrolytes [...]
> > [...] empirical validation is primarily confined to simple metallic systems (FCC copper)
>
> Our main body evaluation runs three systems: FCC copper, bulk water, and an LGPS solid electroyte. Yet, we do agree that the rejection rate scaling should be tested on more systems than copper. Hence, we ran new comparisons of observed rejection rates with theory (our scaling equation 12), with and without EC, on **four alanine oligopeptides** as well as a more complex bulk system (our **LGPS electrolyte**). The empirical constant $\varepsilon$ is averaged across its estimates from all atom counts for a more robust fit.
>
> ## Alanine oligopeptides ($300 \text{K}$, $\gamma = 5\text{ps}^{-1}$)
>
> ### No error correction
> |System|Atom count|Rejection rate [%]|Theory [%]|
> |-|-|-|-|
> |ala2|22|4.5 ± 0.8|5.6|
> |ala4|42|7.3 ± 0.5|7.7|
> |ala6|62|10.4 ± 0.9|9.3|
> |ala8|82|12.2 ± 0.5|10.7|
>
> ### With error correction
> |System|Atom count|Rejection rate [%]|Theory [%]|
> |-|-|-|-|
> |ala2|22|3.5 ± 0.5|4.2|
> |ala4|42|5.9 ± 0.4|5.8|
> |ala6|62|7.6 ± 1.0|7.1|
> |ala8|82|8.8 ± 1.1|8.1|
>
> ## LGPS ($1200 \text{K}$, $\gamma = 0.2\text{ps}^{-1}$)
> ### No error correction
> |Atom count|Rejection rate [%]|Theory [%]|
> |-|-|-|
> |200|41.2 ± 2.1|43.5|
> |400|57.7 ± 2.8|58.5|
> |600|69.8 ± 1.3|68.2|
> |800|77.0 ± 1.5|75.1|
>
> ### With error correction
> |Atom count|Rejection rate [%]|Theory [%]|
> |-|-|-|
> |200|11.4 ± 1.2|11.0|
> |400|15.9 ± 0.7|15.6|
> |600|18.9 ± 1.7|19.0|
> |800|20.8 ± 0.7|21.9|
>
> We observe that:
> - EC improves rejection rates **across the board**. The improvements are drastic for LGPS, where EC is *required* to even fundamentally access the low-$\gamma$ regime (in which Li ions can diffuse freely) at practical rejection rates.
> - For the alanine peptides, the scaling fit systematically overestimates rejection rates at low, and undererstimates them at high atom counts – i.e., rejections unfortunately **grow somewhat faster** than theory would predict. This indeed points at a partial breakdown of the homogeneity assumption.
>
> Still, the theory models a **correct qualitative trend**, and the seen increase in rejections is slow enough to warrant a **practical size regime**.
>
> Thank you for suggesting this experimental addition. We hope it shows that LSD, despite its early stage, is already scalable enough to be useful on systems beyond copper.
>
> > Is there any risk of error accumulation or oscillatory behavior over long trajectories?
>
> This question foresees an issue we had in early testing of EC. Our first approach used the last-known force error of the *EC-corrected* draft prediction to offset the current draft prediction, which led to recursive self-feedback causing instability. Practically, we solved this by computing offsets from the *EC-uncorrected* draft prediction instead. After this change, we could not observe any single EC failure in our experiments.
>
> > Could you provide a more detailed algorithmic description of EC (perhaps as pseudocode)
>
> We plan to publish EC pseudocode in the appendix additions on our async protocol, see our answer to reviewer 5Ny5.
>
> TL;DR: when step is verified without pending precursors, it is appended to a `verified` queue. This queue is emptied and its content commited to the integrator before making a new draft call. The final step thus taken from `verified` is what we mean by the "last known verified step". This step stores both the target and the EC-uncorrected draft forces and we simply add the error between them to the next force prediction made by the draft.
>
> > The "fixed compute overhead" cited for GP may be implementation-specific and potentially optimizable
>
> We show to reviewer AF73 that this fixed overhead is in fact a general feature shared across current SOTA MLIP architectures, giving LSD a unique advantage over spatial parallelism in the ~1000-atom regime.
>
> > The paper should frame this trade-off more transparently as a size-dependent limitation
>
> We currently try communicating this in ll. 256-263 / 370-375 but would be glad to take additional feedback.
>
> > Can you sketch how LSD would be applied to a BAOAB integrator?
>
> Thank you for pointing us to this. First, note that generic LSD only requires *some* maximal coupling to be known, which might independently exist for BAOAB. Yet, *our* practical template to construct such couplings does not transfer (no contiguous block that could be reflection-coupled).
>
> However, it *does* transfer to OBABO, which has the same superconvergent properties as BAOAB and is optimal for momentum-based observables (see *arXiv: 1203.5428*). We need apply the coupling maps staggered by a half-step, i.e., OBA(BOOB)A(BOOB)A[...]. When a staggered step is accepted, we can commit an un-staggered sample to the simulation by using the target forces available post-verification. We will add details in a separate appendix section on this topic.

---

> > ### Author Rebuttal · Reviewer_JcAk · 2026-04-03
> >
> > It is impressive to see the new numerics which do indeed help to convince.
> >
> > One small note: ``...OBABO, which has the same superconvergent properties as BAOAB''  - that is not actually a correct statement.

---

> > > ### Author Response · Authors · 2026-04-03
> > >
> > > We are happy to hear that you find our numerical results convincing. We have just discovered one error in our rebuttal reply, and want to correct it in this edit: the LGPS rejection rate values were taken for our default setting of $\gamma = 1 \text{ps}^{-1}$ – we forgot to lower it to the $\gamma = 0.2 \text{ps}^{-1}$ value used for our main body diffusion coefficient experiment, which makes rejections overall lower compared to $\gamma = 0.2 \text{ps}^{-1}$. We will set this right in our camera-ready additions and hope that, also after this correction, you still feel that we could address your primary concern about transferability of the rejection rate scaling model.
> > >
> > > *Edit 04/04/2026*: We have now obtained LGPS rejection rates with the lowered value of $\gamma = 0.2 \text{ps}^{-1}$ that we used in the main body diffusion coefficient experiment.
> > > - As expected from theory, rejection rates are overall higher due to this extremely low-friction (i.e. low-noise) setting. However their size scaling still follows the theoretically expected trend.
> > > - the effect of EC is even more pronounced.
> > >
> > > We believe that absolute rejection rates can be further lowered by future draft-target alignment efforts beyond just our use of EC, e.g. teacher-student distillation.
> > >
> > > ### LGPS at $\gamma = 0.2\text{ps}^{-1}$ — no error correction
> > >
> > > | Atom count | Rejection rate [%] | Theory [%] |
> > > |------------|--------------------|------------|
> > > | 200 | 77.7 ± 1.3 | 80.8 |
> > > | 400 | 92.8 ± 1.3 | 93.5 |
> > > | 600 | 97.9 ± 0.4 | 97.6 |
> > > | 800 | 99.5 ± 0.3 | 99.1 |
> > >
> > > ### LGPS at $\gamma = 0.2\text{ps}^{-1}$ — with error correction
> > >
> > > | Atom count | Rejection rate [%] | Theory [%] |
> > > |------------|--------------------|------------|
> > > | 200 | 18.0 ± 1.0 | 16.7 |
> > > | 400 | 23.1 ± 0.9 | 23.5 |
> > > | 600 | 27.6 ± 1.2 | 28.6 |
> > > | 800 | 32.1 ± 0.8 | 32.8 |
> > >
> > > Regarding your note: we just had another careful look at the asymptotic analysis in [1] and can confirm that the claim was a mistake from our side. Thanks for the great catch! We had derived this mistake from looking at [2], Fig. 4(a) where OBABO (called BP in this work) is the only integrator seen to outperform 2nd-order accuracy when kinetic (not configurational) temperatures are recorded. We mistook this specific result to be a consequence of a general superconvergence property of OBABO for momentum observables.
> > >
> > > Let us end on three notes that might perhaps address your remaining concern.
> > > 1. A **specificity to particular splitting choices**, and particularly the preference for OBABO over BAOAB, is **not unseen in prior methods**. For instance, FlashMD ([3], NeurIPS 2025 spotlight) can similarly be adapted to OBABO but not BAOAB (cf. page 23). This relates to the fact that OBABO encloses a VV block, a technicality not unlike our need for a "contiguous momentum block". Likewise, there is a preference for OBABO in HMC work [4].
> > > 2. At least from the viewpoint of theory, the superconvergence advantage of BAOAB over other splits holds **only in the high-$\gamma$** (i.e., high-friction) regime [1,2]. However, as ballistic propagation is impacted in this regime, it is anyways not a practical setting if one aims to faithfully sample kinetic observables.
> > > 3. **Also empirically** ([2], Fig. 5), an advantage of BAOAB over ABOBA is no longer obvious for frictions below $1\text{ps}^{-1}$ and timesteps below $1.5 \text{fs}$, a timestep that is in practice rarely exceeded for unconstrained simulations.
> > >
> > > As this is the final comment we can make, we thank you again for your constructive feedback and your great suggestion of added rejection rate scaling experiments.
> > >
> > > [1] **Benedict Leimkuhler, Charles Matthews**; Rational Construction of Stochastic Numerical Methods for Molecular Sampling, Applied Mathematics Research eXpress, Volume 2013, Issue 1, 2013, Pages 34–56, https://doi.org/10.1093/amrx/abs010. **arXiv:1203.5428**.
> > >
> > > [2] **Benedict Leimkuhler, Charles Matthews**; Robust and efficient configurational molecular sampling via Langevin dynamics. J. Chem. Phys. 7 May 2013; 138 (17): 174102. https://doi.org/10.1063/1.4802990. **arXiv:1304.3269**.
> > >
> > > [3] **Filippo Bigi, Sanggyu Chong, Agustinus Kristiadi, Michele Ceriotti**; FlashMD: long-stride, universal prediction of molecular dynamics. The Thirty-ninth Annual Conference on Neural Information Processing Systems (NeurIPS 2025). **URL: https://openreview.net/forum?id=ogZu06NgQs**.
> > >
> > > [4]  **Pierre Monmarché**; High-dimensional MCMC with a standard splitting scheme for the underdamped Langevin diffusion. Electronic Journal of Statistics Vol. 15 (2021) 4117–4166. ISSN: 1935-7524. https://doi.org/10.1214/21-EJS1888. **arXiv: 2007.05455**.

---

### Official Review · Reviewer_5Ny5 · 2026-03-12

**Soundness:** 2
**Presentation:** 2
**Significance:** 3
**Originality:** 3
**Overall Recommendation:** 4
**Confidence:** 2

**Summary:**

This paper introduces Langevin Speculative Dynamics (LSD), a speculative sampling framework designed to accelerate molecular dynamics (MD) simulations. The central idea is to employ a fast, inexpensive draft model to propose candidate integration steps and a slower, more accurate target model to verify them in parallel. The key technical contribution is an adaptation of speculative verification from first-order sampling to the second-order Langevin dynamics relevant to MD, achieved via a reflection-maximal coupling for the momentum update and a pre/post-processing argument that lifts the coupling to full integration steps. The paper further develops a speedup analysis, a rejection-rate scaling model, and a speculative error-correction heuristic. Experiments on copper, bulk water, and LGPS systems report substantial wall-clock gains in favorable regimes, while several physical observables appear to remain close to those of the target simulator.

**Compliance With Llm Reviewing Policy:**

Affirmed.

**Final Justification:**

The authors have addressed major concerns

**Key Questions For Authors:**

1.	Proposition 3.1 does not appear to follow from its stated assumptions: the Jensen step yields the wrong direction for a concave $f$, and using an upper bound $r$ on the rejection probability adds on to this issue. Can you provide a corrected statement and proof, and clarify whether the intended result is a rigorous bound or a heuristic expression in terms of a fixed true rejection rate?
2.	Eq. (2) writes the O-step noise as $\Sigma,\xi$, but later equations treat $\Sigma$ as a covariance and operate on $\Sigma^{1/2}$. Relatedly, Eq. (19) in Appendix C uses $M^{-1}\Delta F$ where the Mahalanobis distance appears to require $M^{-1/2}\Delta F$. Can you state the intended conventions explicitly and correct the derivations where needed?
3.	The one-step coupling argument is clear, but the headline claim is about the full asynchronous pipeline with rollback and potentially out-of-order verifier returns. Can you give an end-to-end exactness argument for this setting in the main paper?
4.	Speedup is reported under varying GPU counts across configurations, which makes the $3\times$-$9\times$ headline numbers hard to evaluate as a systems contribution. Can you either normalize by GPU-hours or compare under a fixed hardware budget?

My main concern is the correctness and presentation of the theory. If the rebuttal convincingly resolves the issues around Proposition 3.1, the stochastic notation, and the exactness argument for the pipelined implementation, I would be open to updating my assessment.

**Limitations:**

yes

**Strengths And Weaknesses:**

Strengths:

The paper addresses an important and technically interesting problem. Molecular dynamics is intrinsically sequential, and this serial bottleneck becomes especially costly when the target force model is expensive. The idea of pairing a cheap draft model with speculative verification is well motivated and potentially impactful for ML-driven atomistic simulation. The core construction is well-crafted: in particular, the reduction from full-step verification to a Gaussian coupling on the momentum update is conceptually clean, and the reflection-based verification mechanism is one of the strongest aspects of the paper. The main intuition is also communicated effectively through the figures, especially the overview of the asynchronous pipeline and the illustration of the reflection coupling. The target-parity checks on temperature and diffusivity serve as useful sanity tests, and the comparison against graph parallelism is informative for delineating the practical regime in which the method may be attractive. I also appreciate that the paper does not rely solely on headline speedup numbers. The discussion of rejection rates, cost ratios, and the cost-versus-rejection tradeoff provides readers with useful intuition about when the method is and is not likely to be beneficial.

Weaknesses:

1.	Proposition 3.1 appears to be false as stated. The proof applies Jensen's inequality to $f(x) = \frac{1}{c + x^{-1}} = \frac{x}{cx+1}$, but $f''(x) = -\frac{2c}{(cx+1)^3} < 0$, so $f$ is concave and Jensen gives $\mathbb{E}[f(L)] \leq f(\mathbb{E}[L])$, not equality. Moreover, if $r$ is only an upper bound on the rejection probability, then $\mathbb{E}[L] \geq 1/r$, and since $f$ is increasing, $f(\mathbb{E}[L]) \geq f(1/r) = \frac{1}{c+r}$-the opposite direction from the claimed upper bound. A concrete counterexample confirms this: with $c=0.1$, true rejection rate $\beta=0.05$, and upper bound $r=0.2$, the expected speedup under the paper's own streak model is approximately $5.47$, which strictly exceeds $1/(c+r)=3.33$. The proposition is therefore not a valid upper bound.

2.	There is a notation and formula inconsistency in the stochastic integrator. Eq. (2) writes the O-step as $p \leftarrow e^{-\gamma \Delta t}, p + \Sigma, \xi$, while Eq. (3) defines $\Sigma$ as a covariance-like quantity. Later equations-particularly the reflection-coupling construction in Eqs. (8)-(10)-treat $\Sigma$ as the covariance matrix and operate on $\Sigma^{1/2}$ and $\Sigma^{-1/2}$. If $\Sigma$ denotes the covariance, then Eq. (2) should read $\Sigma^{1/2}\xi$, not $\Sigma,\xi$. The printed formulas as written are incorrect and directly affect the derivation in Appendix C.

3.	The rejection-rate derivation in Appendix C contains a mathematical error at a key step. From the paper's own definitions, the Mahalanobis distance should scale with $|M^{-1/2},\Delta F|$, whereas Eq. (19) uses $M^{-1},\Delta F$. This means the derivation of the scaling law does not follow from the earlier equations as claimed. While the final scaling form may retain heuristic value, Eqs. (19)/(20) should be presented as a semi-empirical hypothesis rather than a clean derivation.

4.	The exactness guarantee for the full asynchronous pipeline is insufficiently developed in the main paper. The one-step coupling construction and the rollback intuition are plausible: once a rejection occurs, stale steps are flushed and simulation restarts from the last verified state, so the retained prefix should follow the correct target conditional. However, the paper's headline claim concerns the end-to-end asynchronous multi-device procedure, and Appendix D explicitly states that Algorithm 2 is a simplification that omits out-of-order target returns.

5.	The empirical evaluation is too narrow relative to the breadth of the claims. Most of the analysis is internal: draft vs. target vs. LSD, with one additional graph-parallel comparison. This suffices to demonstrate that the idea can work, but it is not enough to situate LSD among the broader landscape of acceleration strategies discussed in the paper. At minimum, the paper should either include a stronger external baseline or explain more explicitly why such a comparison would be scientifically mismatched.

---

> ### Author Rebuttal · Authors · 2026-03-31
>
> We thank you for cross-checking our theoretical content in depth. Your feedback has helpfully uncovered several errors in our previous draft. We aim to clear up any potentially caused confusion in this rebuttal and to convince you that none of these mistakes affect our major claims.
>
> ### A) Corrections to the Manuscript
> 1. Corrections regarding Prop. 3.1:
> - The second proof equality should indeed be an inequality. Thank you for spotting this (we did intend to typeset an inequality here, which is why we referred to Jensen).
>
> - The assumption should state "let $r > 0$ lower-bound [...]" instead of "upper-bound", as finding an anywhere lower (i.e. better) rejection rate bounds the speedup from above. Both inequalities then point in the right direction, resulting in an overall upper bound.
>
> - This does not influence our derivation of Eq. 11 and follow-up analysis in Sec. 3.3 as they anyway assume the simpler model of I.I.D. rejections at a constant "time-averaged" rate $\langle \beta \rangle$.
>
> 2. $\mathbf{\Sigma}$ should denote a covariance matrix, so the correct O-step in Eq. 2 should indeed be $\mathbf p \leftarrow e^{-\gamma \Delta t} \mathbf p + \mathbf{\Sigma}^{\frac{1}{2}}\mathbf{\xi}$, we apologize for this typo. The later usage of $\mathbf{\Sigma}$ is correct.
>
> 3. $\mathbf{M}^{-1}$ needs to be replaced by $\mathbf{M}^{-\frac{1}{2}}$ in App. C. This does however not affect our use of Eq. 12 in the main body: the scaling ansatz still works with the corrected mass factor and the per-atom error norm is still absorbed into $\varepsilon$.
>  > Eqs. (19)/(20) should be presented as a semi-empirical hypothesis rather than a clean derivation.
>
> Eq. 19 is first-order exact. The scaling ansatz leading to Eq. 20 (with the corrected $\mathbf{M}^{-\frac{1}{2}}$) is indeed a semi-empirical assumption, which we lay out as Assumption 2. We do intend to make this very clear to the reader by our own wording of "semi-empirical relationship" (l. 243) and our cautioning remarks in ll. 266-271.
>
> ### B) Correctness of the asynchronous pipeline
> We agree that this should be reconstructible from the paper, not just our source code, and will add a formal treatment based on the following remarks to App. D of the paper.
>
> An async protocol must implement the following outside contract to ensure correctness: for an initial condition $x_0$, it must sample an ordered sequence of `verified` MD time steps $x_1, \dots, x_n$. `verified` is defined recursively as:
> - $x_0$ is `verified`, and
> - $x_i$ is `verified` if we can certify that $x_i \sim P(x_i \, | \, x_{i-1})$, where $x_{i-1}$ is `verified` and $P$ is the target conditional distribution for a single MD step (we overload sample vs. random variable notation for simplicity).
>
> Assuming this holds, correctness of the joint distribution follows inductively as in ll. 154-164.
>
> To maintain the contract under async and (possibly) out-of-order target returns, all draft steps in the pipeline go through internal book-keeping.
> - A new draft step gets a unique, immutable, ascending integer `id`.
> - When submitting it to a target model for verification, its `id` gets registered as `eligible_pending`.
> - All steps returned by a target model went through reflection coupling, i.e. are either accepted, or rejected and overridden. We listen for newly returned steps after every draft call and handle each of them as follows:
>      - Regardless of acceptance or rejection, the `id` is moved to `conditionally_verified` if it is still in `eligible_pending`. Else, abort to handle other newly returned steps.
>      - If the step was rejected, deregister all larger `id`s from `eligible_pending` and `conditionally_verified`.
>      - Check if any `id` in `eligible_pending` is smaller than the minimum `id` in `conditionally_verified`. If not, pop the minimum `id` and append its indexed step to a `verified` queue. Repeat until no more new steps can be added to `verified`, then proceed to handle other newly returned steps.
>
> We see that `conditionally_verified` caches potential OOO returns, and that an MD step is added to `verified` iff:
> - It was `conditionally_verified`,
> - All earlier steps were `conditionally_verified` and none of them was rejected.
>
> The first condition implies that the step went through a reflection coupling, while the second one implies that its predecessor was added to `verified`. Therefore, "added to `verified`" satisfies the recursive contract that ensures the correct joint distribution.
>
> ### C) Extended baselines and GPU counts
> Our results in response to reviewer Af73 generalize our observed notion of a "graph parallelism ceiling" to different models as their minimum latencies fundamentally bottleneck *any spatial* parallelism code. This shows the unique strength of speculative over spatial parallelism for low (~1000) atom counts. We show that this minimum latency ceiling is broken by LSD also at fixed, (suboptimally) lowered GPU counts.

---

> > ### Author Rebuttal · Reviewer_5Ny5 · 2026-04-02
> >
> > I increased to 4

---

> > > ### Author Response · Authors · 2026-04-04
> > >
> > > Dear Reviewer 5Ny5,
> > >
> > > We are truly grateful for your careful checking and constructive feedback that helped us weed out multiple mistakes in our draft. Our formal argument for correctness of the asynchronous protocol is also a great addition that will surely be helpful to many readers. Thank you likewise for appreciating our revisions by a raising of your score.
> > >
> > > Warmest regards,
> > >
> > > Authors of Submission #17654

---

### Official Review · Reviewer_x1ZY · 2026-03-12

**Soundness:** 3
**Presentation:** 3
**Significance:** 3
**Originality:** 3
**Overall Recommendation:** 5
**Confidence:** 3

**Summary:**

The authors propose Langevin Speculative Dynamics (LSD), generalizing previously proposed speculative sampling to second-order Langevin sampling that allows usage for running molecular dynamics simulations. A fast draft model proposes a stream of steps, which are verified by a number of parallel target models. LSD preserves the target distribution and yields 3-9x speedups in the settings tested by the authors. The authors further provide theoretical insights into the dependency of the speedup on many parameters of the simulation setup.

**Compliance With Llm Reviewing Policy:**

Affirmed.

**Final Justification:**

The authors addressed my concerns, particularly regarding the practical use cases of the methodology and the requirements that the draft needs to fulfill. I thus raised my overall recommendation to 5.

**Key Questions For Authors:**

5. I wonder what the efficiency gain from the asynchronous pipeline is compared to a synchronous approach. Is it possible to approximate its impact?
6. The authors note that LSD works with “arbitrary draft models.” I assume this is only true under the assumption that the draft model sufficiently covers the relevant parts of the state space with respect to the target distribution. Can the authors expand on this point? I suggest adding a targeted discussion of what a draft model needs to look like in practice. The current claim that LSD works with “arbitrary draft models” appears too strong.

**Limitations:**

A key limitation of the approach is its scalability to large systems, where rejection rates increase significantly. However, the authors explicitly acknowledge this issue, and it does not diminish the usefulness of the method, as there appears to be a reasonable regime in which it can be applied effectively.

**Strengths And Weaknesses:**

**Strengths**

- Strong empirical speedups observed.
- The authors provide a useful theoretical analysis of the achievable speedup as a function of simulation setup parameters, which is in good agreement with the experiments shown.
- Instead of a synchronous rollout of draft steps and subsequent verification, the shown algorithm is asynchronous, which yields additional efficiency.

**Weaknesses**

1. The authors state that arbitrary draft models can be used. While this might be true in theory, there must be practical limitations that, in my view, are not discussed sufficiently (see my question below).
2. The authors check for preservation of the target distribution by evaluating self-diffusivities for lithium diffusion and by computing the MMD over short trajectories. I do not find these tests convincing for demonstrating that the proposed sampling scheme actually preserves sampling from the correct target Boltzmann distribution. At the very least, quantities such as radial distribution functions, or similar measures, should be evaluated to assess the correctness of the distribution for the test systems.
3. More generally, the test systems (mainly FCC copper and bulk water) are very limited. These systems are suitable for evaluating the scaling and stability of the method, but I do not think they are sufficient to demonstrate that the proposed approach preserves the target distribution for general MLIP sampling. Including a system such as a small peptide and showing that LSD preserves macroscopic observables, for example a Ramachandran plot, would be much more convincing.
   - I acknowledge that the main contribution of the present work lies more on the development and theoretical side. Furthermore, testing preservation of the Boltzmann distribution for more complex systems may be somewhat more challenging than for the systems evaluated in the current manuscript. However, without such tests, many of the claims regarding distributional preservation should be significantly toned down, and at the very least an extended discussion on how correct sampling could be evaluated for more complex systems is necessary.

**Other comments**

4. The reported speedups appear to be calculated with respect to a single serial instance of the target model. Since the proposed method uses multiple target verification models in parallel, this comparison is at least somewhat questionable. The chosen baseline seems fair only when a single consecutive trajectory needs to be sampled and running multiple trajectories in parallel is not beneficial. While such cases do exist, this is often not a realistic setup, as running parallel trajectories is very frequently possible. An extended discussion of this point is needed so that readers understand in which situations the proposed method is useful.

---

> ### Author Rebuttal · Authors · 2026-03-31
>
> Thank you for appreciating our empirical results, theoretical speedup analysis and asynchronous pipelining idea.
>
> ### Scope of mathematical guarantees
> We appreciate your feedback from W1 / Q6 – let us clarify that our notion of "arbitrary draft model" is, up to machine precision, **not an oversimplification**. We will append this paragraph to Sec. 3.1:
>
> > The LSD method in its ABOBA-specific form therefore provides the following parity guarantees. Let $F, \tilde F \colon \mathbb{R}^{3N} \rightarrow \mathbb{R}^{3N}$ be the draft and target force field for atom count $N$. For all trajectory lengths $n$ and initial conditions $x_0 = (\mathbf{q}_0, \mathbf{p}_0)$, LSD samples from the joint target model distribution $P(x_1, \dots, x_n \, | \, x_0)$ of ABOBA trajectories generated through (Eq. 2) from the target force field $F$, entirely regardless (!) of the functional form of $\tilde F$, up to artifacts of finite machine precision.
>
> This very strong guarantee is true because the coupling property holds for reflection couplings (Eq. 9) regardless of the separation between Gaussian means, and therefore regardless of the separation between draft and target forces.
>
> Of course, while every draft is *valid* in terms of parity guarantees, not every draft is also *useful* in terms of speedups determined by the rejection rate, i.e. Gaussian overlap.
>
> ### Further parity experiments: matching LSD / target radial distribution functions and Ramachandran plots
> While any correct LSD implementation must match the above guarantees, and therefore *all* its target model observables by design, we definitely agree that more parity experiments are of good use, e.g. to check against numerical artifacts.
>
> Following your suggestion, we computed the O-O and H-H radial distribution functions for the same bulk water system and `UMA-S` target + `UMA-tiny-omol` draft setup as studied in the main body. Our result shows that the target model and LSD graphs match up perfectly, while the draft model graph differs significantly (anonymized private link: https://figshare.com/s/5699e7550234476b97d8). This is consistent with our strong mathematical guarantees.
>
> We also started gathering Ramachandran plots for alanine dipeptide and will share them when converged.
>
> ### Quantifying efficiency impact of asynchronous pipelining
>
> Thanks to your suggestion. The speedup of the synchronous setup (no pipelining) is derived by [1] (Theorem 3.8) under an identical assumption as Eq. 11 (i.i.d. Bernoulli rejections). Unlike our Eq. 11, the synchronous speedup factor $F_\text{sync}$ has a strict dependence on the number of employed target models (i.e. GPUs) $\gamma$. Adapted to our notation with time-averaged acceptance rate $\langle \alpha \rangle$ and cost fraction $c$, it reads
> $$F_\text{sync} = \frac{1+\langle \alpha \rangle ^{\gamma+1}}{(1-\langle \alpha \rangle)(\gamma c + 1)}$$
> while our Eq. 11 (after inserting $\langle \beta \rangle = 1 - \langle \alpha \rangle$) has the simpler form
> $$F_\text{async} = \frac{1}{1-\langle \alpha \rangle + c}.$$
>
> One apparent difference is that $F_\text{sync}$ can never tightly reach the bare draft-target speedup $\frac{1}{c}$ even for perfect acceptance rates $\langle \alpha \rangle \rightarrow 1$ in the nontrivial case of $c < 1$. This is expected, as the synchronous pipeline can only run at draft throughput for bursts of $\gamma$ steps, after which the draft model has to pause for the duration of one (synchronous) verification call to all target models. Meanwhile, pipelined LSD can indeed run indefinitely at full draft throughput (i.e., at speedup $\frac{1}{c}$) when $\langle \alpha \rangle \rightarrow 1$.
>
> Formally, the statement just made follows from $$\frac{1-\langle \alpha \rangle^{\gamma+1}}{(1-\langle \alpha \rangle)(\gamma c+1)} = \frac{\sum_{k=0}^{\gamma}\langle \alpha \rangle ^k}{\gamma c+1} \leq \frac{\gamma +1}{\gamma c+1} < \frac{1}{c}$$
> where we used the truncated geometric series expansion, followed by $\langle \alpha \rangle \leq 1$, followed by $c < 1$ for the final, strict inequality.
>
> We think that this quantitative analysis of pipelining is a great addition to our work and will therefore include it in Appendix D.
>
> ### Comparison to batch parallelism
>
> > The chosen baseline seems fair only when a single consecutive trajectory needs to be sampled and running multiple trajectories in parallel is not beneficial. While such cases do exist, this is often not a realistic setup, as running parallel trajectories is very frequently possible.
>
> We mention in our conclusion (ll. 426-428) that
>
> > [...] LSD focuses on throughput under the assumption of sufficient compute, which may not always be practical.
>
> What we mean by this is that, given a fixed compute budget, exhausting more obvious forms of parallelism first is potentially more useful. However, if compute is still available, LSD is not a conflicting option, but can be used to further speed up each parallel trajectory.

---

> > ### Author Rebuttal · Reviewer_x1ZY · 2026-04-01
> >
> > I thank the authors for their comments and answers to my questions.
> >
> > Their answer addresses my concerns partially, though I have remaining questions / concerns that need to be addressed:
> > - "Of course, while every draft is valid in terms of parity guarantees, not every draft is also useful in terms of speedups determined by the rejection rate, i.e. Gaussian overlap."
> >   - I think this point is very important. For example, when the draft model has zero probability of sampling a part of the phase space, LSD will never reach this part either, even when it has a high probability in the target distribution. While mathematically the authors may be correct about the given guarantees, it is not clearly communicated what this means in practice. The limitations are not adequately discussed, even though they are very important for practitioners that want to apply such a method in practice.
> > - "What we mean by this is that, given a fixed compute budget, exhausting more obvious forms of parallelism first is potentially more useful. However, if compute is still available, LSD is not a conflicting option, but can be used to further speed up each parallel trajectory."
> >   - Given this statement, even if sufficient compute is available, using "more obvious forms of parallelism" such as simply running more trajectories on the same amount of compute might still be better. It sounds to me like the number of scenarios where LSD is useful in practice is very limited. An extended discussion on the scenarios where LSD is actually beneficial is crucial, and I'm currently not convinced by the discussion provided by the authors on this matter.
> >
> > **edit:**
> > I thank the authors for the clarifications below. I did indeed misunderstand parts of the methodology, since I previously believed rejected samples to be entirely discarded. I thank the authors for this important clarification. Looking at the manuscript again, this is definitely implicitly stated; however, I still suggest explaining this important fact somewhere more explicitly.
> >
> > I further appreciate the authors discussing the practical application more clearly, specifically that LSD is mainly useful when long continuous trajectories are needed. I suggest highlighting this very prominently in the manuscript, as this is an important information for practitioners.
> >
> > My concerns are now addressed, and I will update my score accordingly.

---

> > > ### Author Response · Authors · 2026-04-02
> > >
> > > We appreciate that you took the time to engage with our rebuttal early in the discussion period! We hope our responses adequately address all your concerns.
> > >
> > > > "when the draft model has zero probability of sampling a part of the phase space, LSD will never reach this part either, even when it has a high probability in the target distribution."
> > >
> > > This is a great edge case and helps us clarify something important and easily misunderstood: LSD **requires no draft coverage** of target-sampled regions to ensure exact target sampling. This is possible because, unlike e.g. standard rejection sampling, LSD is a coupling-based method that **actively transforms and then commits rejected samples** instead of just discarding them.
> > >
> > > Here is precisely how exact sampling works in your edge case:
> > >
> > > - All samples get rejected. The reflection coupling then transports this sample population from the draft- into the target-sampled region and recovers the exact target model density.
> > > - As the target model needs to override every draft sample, the resulting inference time per step is one draft + one target model call (meaning no speedup but actually a slight slowdown).
> > >
> > > For a visual illustration of this zero-overlap case, it helps to imagine Fig. 2 (a)-\(c) with two very distant Gaussians. In this case, the green area vanishes and the red area of rejected samples becomes the full left Gaussian. It then gets reflected over to the right-hand side which maps it exactly onto the target Gaussian.
> > >
> > > > The limitations are not adequately discussed [...]
> > >
> > > The exact sampling guarantee is not a theoretical approximation with limited practical scope, but is **precisely how LSD should and does behave in practice**. There are no extra assumptions that could be violated in practice. Our four parity results,
> > > - indistinguishably matching water RDF curves from rebuttal
> > > - matching diffusion coefficients
> > > - elimination of temperature drift
> > > - vanishing MMD kernel distances
> > >
> > > all serve to confirm that our LSD implementation is correct and does what it should do **by construction**.
> > >
> > > We hope that this clarifies our view on why there is in fact no undiscussed accuracy limitation. LSD sampling accuracy is 100% lossless in both theory and practice.
> > >
> > > > It sounds to me like the number of scenarios where LSD is useful in practice is very limited.
> > > > An extended discussion on the scenarios where LSD is actually beneficial is crucial
> > >
> > > We had to keep our initial response shorter than we would have liked and apologize if this led to confusion. To clarify:
> > >
> > > 1. LSD and trajectory-level parallelism can always be **trivially combined** by accelerating each parallel trajectory with LSD. Given sufficient compute, they are **not at all exclusory**.
> > > 2. Assuming fixed compute, one can **sometimes** "trade" LSD against trajectory-level parallelism, but whether this is possible is **highly problem-dependent**.
> > >
> > > Plenty of interesting MD problems intrinsically rely on long, contiguous trajectories and cannot just be solved by running many short and parallel ones instead. The existence of such intrinsically serial problems is naturally a main reason why the long-timescale bottleneck of MD is still considered far from solved. We agree, however, that our work should better pinpoint the scope of such problems and will add this to our introduction section:
> > >
> > > > Examples of problems requiring long time scales include long-time kinetic quantities (transition rates, mean first passage times), transport properties computed from time-correlation functions, or pathway discovery where temporal ordering is essential. In all these cases, the binding constraint is the wallclock time of individual continuous trajectories, precisely the performance axis that LSD targets.
> > >
> > > In summary, and also relating to parts of our response to reviewer Af73, LSD is most useful *at fixed compute* if:
> > >
> > > 1. The problem intrinsically requires long sequential simulation trajectories, as opposed to just many parallel short trajectories (-> regime of advantage against trajectory-level parallelism)
> > > 2. System sizes are moderate (up to ~1000-2000 atoms, see our response to reviewer Af73 -> regime of advantage against spatial parallelism)
> > > 3. A Langevin thermostat is desired
> > >
> > > We will add this more precise characterization to the limitations part of our discussion section.
> > >
> > > LSD adds a **genuinely novel form of parallelism** to the MD toolbox that has unique tradeoffs against both spatial and trajectory-level parallelism, so we agree that it is extremely important to transparently place it in the existing landscape. When it is used under the right conditions, we think LSD can push the boundaries of the absolute time scales accessible with accurate MLIPs.
> > >
> > > Hence, thank you for your feedback so far! It helped us develop a more precise characterization of LSD usefulness, which we hope is now more clearly communicated through the proposed revisions.

---

### Official Review · Reviewer_Af73 · 2026-03-13

**Soundness:** 3
**Presentation:** 3
**Significance:** 2
**Originality:** 4
**Overall Recommendation:** 4
**Confidence:** 4

**Summary:**

The authors introduce Langevin Speculative Dynamics (LSD), porting over ideas from speculative sampling for diffusion models to molecular dynamics. LSD provably preserves the distribution of the slower, target model while increasing speedup of overall simulation time via a draft model and a verification step.

**Compliance With Llm Reviewing Policy:**

Affirmed.

**Final Justification:**

The rebuttal has addressed my concerns. LSD, at the moment, doesn't seem to be practically useful outside of a specific use case (Langevin NVT, many many GPUs, small system sizes). However, the paper is written well and is technically sound. Furthermore, the idea of LSD is novel and surprising. Taking this altogether, I increase my score to a 4 but decrease confidence from 5 to 4.

**Key Questions For Authors:**

See above.

**Limitations:**

yes

**Strengths And Weaknesses:**

The method seems rather limited to a Langevin thermostat operating in NVT. Am I correct in noticing that the speculating sampling approach presented in the work does not extend to widely used configurations such as NVE, NVT with other thermostats such as Nose-Hoover, or NPT ensembles?

Does focusing on other splits instead of ABOBA affect the results? Are there advantages in using other splits for real world MD applications? Does the speculative sampling algorithm as proposed naturally extend to other splits?

Can you elaborate on the Figure 5 setup and comparison to graph parallelism? How do speedups associated with LSD compare to approaches like Distmlip [1] or LAMMPS multi-GPU inference [2]? The primary question I ask is: given a fixed amount of GPUs, when should one use LSD and when should one use standard multi-GPU inference? The calculations used to inform Figure 5 are not clear. The “Graph Parallelism Ceiling” also seems specific to the model and implementation used and not specific to graph parallelism as a technique. Furthermore, using 8 or 64 GPUs seems extremely excessive for relatively tiny atomic systems consisting of a couple thousand atoms. The manuscript mentions that “LSD performs an order of magnitude better than graph parallelism for small N and trails GP for large N.” But the crossing point in which GP is better occurs around 2100 atoms. 2100 atoms is a very standard system size for many MD applications. I question the practical applied utility of LSD if speedups relative to GP only appear in system sizes smaller than 2100 atoms.

Furthermore, I highly encourage the authors to benchmark against LAMMPS multi-GPU inference (perhaps with the MACE model due to its accessible LAMMPS implementation) and provide a head to head comparison in speedup against LSD using the same fixed # of GPUs. LAMMPS is known to have a fast multi-GPU implementation and is also extremely widely used in the field for multi-GPU simulations. I believe this should be the ground truth benchmark instead of GP. Otherwise, it’s difficult to trust the results in the manuscript behind the utility of LSD.

Weaknesses in summary:
* Doesn’t seem applicable to popular configurations in MD such as NVE, NVT with other thermostats, or NPT
* Runtime is lower bounded by the runtime of the draft model. Although classical potentials can be used as draft models, the generalizability of the proposed method is then restricted to the chemical space of the draft model.
* LSD allows someone to throw more GPUs at the simulation problem in exchange for faster simulation. There already exist baselines in multi-GPU inference such as LAMMPS multi-GPU that weren't benchmarked against thoroughly.

The paper introduces an interesting theoretical idea in speculative sampling for MD simulation, but the practical utility in generality to different configurations of simulation and comparisons to multi-GPU baselines are unknown. Therefore, I give a score of 3. However, if the authors provide a comparison showing superior performance relative to such baselines across a variety of system configurations, I will increase my score. An analysis on the necessary conditions to use LSD as opposed to LAMMPS would be very interesting as well. I also think increasing the GPU configurations beyond 8 and 64 GPUs for LSD benchmarking would be more comprehensive (such as 2 or 4 GPUs) as many groups running MD don't have access to 64 GPUs.

[1] https://arxiv.org/abs/2506.02023
[2] https://developer.nvidia.com/blog/enabling-scalable-ai-driven-molecular-dynamics-simulations/

---

> ### Author Rebuttal · Authors · 2026-03-31
>
> Thank you for appreciating the originality of our idea. Our submission explores a new kind of parallelism for use in MD workflows: speculative parallelism in time, as opposed to the standard spatial (domain decomposition) parallelism. We believe this work opens up a new avenue to accelerate atomic simulations.
>
> Our rebuttal covers, in order:
> - relevance of the ~1000-atom regime
> - LSD generalizability to any MLIP (no model-specific implementations needed)
> - a more general proof of advantage for small systems
> - extensions to other simulation setups
>
> ## Accelerating the ~1000-atom regime is useful
> It is correct that the strength of LSD parallelism lies in the ~1000-atom regime, where absolute throughputs can surpass graph parallelism by **an order of magnitude**. Many real workflows around inorganic materials, catalysis, or small molecules operate precisely in this small regime and are still often bound by prohibitive simulation times. Especially the AIMD research community has uncovered many productive use cases for simulations in this size regime.
>
> ## Benchmarking against other parallelism implementations
>
> > "How do speedups associated with LSD compare to approaches like Distmlip [1] or LAMMPS multi-GPU inference"
>
> Thank you for this suggestion.
>
> 1. We attempted Distmlip but we found it did not give us a speedup for UMA and the implementation was not able to partition 1000 atoms on more than 4 GPUs.
>
> > 1 GPU(s): speedup: 1.00x
> > 2 GPU(s): speedup: 0.71x
> > 4 GPU(s): Skipped: Partition walls are too close together. See  above message.
> > 8 GPU(s): Skipped: Partition walls are too close together. See above message.
>
> This finding is consistent with the failed 2+ GPU experiments on ~5000 atoms for 3 out of 4 models in the DistMLIP paper, page 19, Fig. 8.
>
> 2. LAMMPs Multi-GPU does not have uniform or easily accessible implementations across MLIPs.
>  * The UMA-lammps implementation uses GP under the hood so it measures nearly the same speed as quoted in the paper
>  * The MACE-lammps multi-gpu (with kokkos) worked for us only with MACE-MP-0, a fast draft-type model similiar to Orb that does not fit into the LSD paradigm (using a fast draft model to speed up a slower, more accurate model). We tried MACE-OMol (a slower and UMA-like model) but its implementation does not work with LAMMPs multi-GPU.
>
> **The distinct advantage of LSD here is it is not bound by custom multi-gpu implementations of the MLIP and can work with any draft-target pair.**
>
> > The “Graph Parallelism Ceiling” also seems specific to the model and implementation [...]
>
> To generalize our finding of a “Graph Parallelism Ceiling” to other MLIPs, we have benchmarked inference times on a minimal two-atom graph for various models (see the table below; to be added into the appendix). These times define minimum latency, and hence the maximum throughput possible under **any multi-GPU implementation based on spatial domain decomposition**,  **on any system**.
>
> |Model|Min Latency on H100 GPU (ms)|
> |-|-|
> |UMA-s compiled|20 ms|
> |MACE-mp (small) cueq|18 ms||
> |MACE-mp (medium) cueq|25 ms|
> |MACE-omol (large) (no cueq)|25 ms|
> |Sevenet Omni FlashTP|20 ms|
> |Orb-v3-**conservative** (note this paper uses Orb-v3-**direct** as draft model) |10 ms
>
> Ceilings at around 20ms are typical with the only exception of Orb-v3-conservative at ~10ms. Hence, most of the modern highly accurate MLIPs cannot run >50 qps, even assuming perfectly linear strong scaling with parallelism. This gives **unequivocal advantage to LSD on small systems** (e.g., 100+ qps at < 500 atoms for EMT draft).
>
> The GP ceiling can also be broken at suboptimally lowered GPU counts (UMA-S+Orb-v3-direct, N=500 FCC Cu, bare UMA-S at 17qps):
>
> |#GPU|3|4|5|6|7|8|
> |-|-|-|-|-|-|-|
> |qps|32|46|58|70|77|78|
>
> Of course, MLIP throughput ceilings and accuracies will continue to improve as the field advances, but so can improvements to LSD with faster or more accurate draft models.
>
> ## Possible LSD extensions
> - It may be possible to overcome the size scaling limitation of LSD in the future with spatio-temporal variants that granularize rejections across partitions of the graph.
> > Am I correct in noticing that [LSD] does not extend to widely used configurations such as NVE, NVT with other thermostats such as Nose-Hoover, or NPT ensembles?
> - LSD couplings make intrinsic use of stochasticity, hence cannot be used with deterministic thermostats and barostats. However, **extensions to stochastic NPT barostats** such as the Langevin piston method [Feller et al., 1995] are possible by defining reflection couplings on the extended Langevin system including the volume pseudovariable. We will discuss this in the appendix. It is also worth noting that frequently used implicit solvent models like GBSA naturally introduce Langevin terms structurally equal to an NVT thermostat.
> > Does focusing on other splits instead of ABOBA affect the results?
> - Extensions to other splittings are possible within bounds, see our answer to reviewer JcAk.

---

> > ### Author Rebuttal · Reviewer_Af73 · 2026-04-02
> >
> > My primary concerns regarding multi-GPU simulation comparisons have been addressed. However, I'm still highly skeptical of the practical utility of LSD for a few reasons:
> > 1. Restriction to the small atom regime: running 8+ GPUs on a system that is only ~1000 atoms seems computationally wasteful, especially for high throughput workflows requiring large amounts of simulations.
> > 2. Restriction to NVT with Langevin thermostat or stochastic NPT barostats. The truly unfortunate thing is that Langevin NVT is highly useful in drug discovery applications -- but those same applications require far more than ~1000 atoms.
> >
> > However, the paper introduces and thoroughly explores a novel and interesting idea -- regardless of the final performance of the system. Draft models will also get faster and more accurate while highly accurate models will undoubtedly get slower and even more accurate. Perhaps in the future, LSD will be a dominant way to perform multi-GPU Langevin NVT. Therefore, I raise my score to a 4.

---

> > > ### Author Response · Authors · 2026-04-04
> > >
> > > Dear Reviewer Af73,
> > >
> > > Thank you so much for recognizing the novelty and future potential of introducing speculative parallelism for MD, as well as for raising your score in response to our additions. We are truly grateful for your constructive feedback and questions, which have for sure helped us pinpoint our current regime of usefulness more precisely for this submission.
> > >
> > > Warmest regards,
> > >
> > > Authors of Submission #17654

---

### Decision · Program_Chairs · 2026-04-30

**Decision:**

Accept (regular)

**Comment:**

This paper considers speculative decoding for molecular dynamics simulations by introducing Langevin Speculative Decoding. This draws from the speculative decoding literature for LLMs and diffusion models where a large model "verifies" the drafts of a smaller model in parallel, leading to significant speedup without any drop in quality. The reviewers are all positive about the impact of this paper for molecular dynamics. Some concerns which were raised were:

1. Limited applicability beyond LVT systems, and scalability beyond 1000 atom systems.
2. Lack of multi-GPU baselines for comparison and correctness of comparisons to baselines.
3. Theoretical mistakes in Proposition 3.1.

These questions were satisfactorily addressed in the rebuttal as per the reviewer acknowledgement. I recommend acceptance.